# Low-degree evidence for computational transition of recovery rate in stochastic block model

**Jingqiu Ding**
ETH Zürich
jingqiu.ding@inf.ethz.ch

**Yiding Hua**
ETH Zürich
yiding.hua@inf.ethz.ch

**Lucas Slot**
ETH Zürich
lucas.slot@inf.ethz.ch

**David Steurer**
ETH Zürich
david.steurer@inf.ethz.ch

## Abstract

We investigate implications of the *(extended) low-degree conjecture* (recently formalized in [MW23]) in the context of the symmetric stochastic block model. Assuming the conjecture holds, we establish that no polynomial-time algorithm can *weakly recover* community labels below the *Kesten-Stigum (KS) threshold*. In particular, we rule out polynomial-time estimators that, with constant probability, achieve $n^{-0.49}$ correlation with the true communities. Whereas, above the KS threshold, polynomial-time algorithms are known to achieve constant correlation with the true communities with high probability [Mas14, AS15].

To our knowledge, we provide the first rigorous evidence for such a sharp transition in recovery rate for polynomial-time algorithms at the KS threshold. Notably, under a stronger version of the low-degree conjecture, our lower bound remains valid even when the number of blocks diverges. Furthermore, our results provide evidence of a computational-to-statistical gap in learning the parameters of stochastic block models.

In contrast, prior work either (i) rules out polynomial-time algorithms with $1 - o(1)$ success probability [Hop18, BBK+21a] under the low-degree conjecture, or (ii) degree-poly($k$) polynomials for learning the stochastic block model [LG24].

For this, we design a hypothesis test which succeeds with constant probability under symmetric stochastic block model, and $1 - o(1)$ probability under the distribution of Erdős Rényi random graphs. Our proof combines low-degree lower bounds from [Hop18, BBK+21a] with graph splitting and cross-validation techniques. In order to rule out general recovery algorithms, we employ the correlation preserving projection method developed in [HS17].

## 1 Introduction

The stochastic block model (SBM) is among the most fundamental models in (social) network analysis and information theory, and has been intensively studied for decades

[HLL83, MNS12, ABH15, KMM$^+$13, Abb18]. A fascinating phenomenon in the SBM is the sharp computational threshold for *weak recovery* of its hidden community structure: efficient algorithms are known for achieving constant correlation with the hidden signal when the signal-to-noise ratio is above a certain threshold [CO10, DKMZ11, Mas14, AS15], while no polynomial-time algorithms have been discovered below this threshold despite significant research effort. This computational threshold is known as the *Kesten-Stigum* threshold (KS threshold) in the statistical physics literature [DKMZ11], and it is an important topic in both probability theory and theoretical computer science.

**Kesten-Stigum threshold in the symmetric stochastic block model.**    For simplicity, we focus on the following special case of the stochastic block model.

**Definition 1.1** (Symmetric $k$-stochastic block model SSBM($n, \frac{d}{n}, \varepsilon, k$))**.** Let $k \in \mathbb{N}^+$ be the number of communities, $d \in \mathbb{N}^+$ be the average degree of the graph, $\varepsilon \in [0, 1]$ be the bias parameter, and $n \in \mathbb{N}^+$ be a multiple of $k$. A graph $Y \in \{0, 1\}^{n \times n}$[1] follows the symmetric $k$-stochastic block model distribution SSBM($n, \frac{d}{n}, \varepsilon, k$) if it is sampled in the following way: assign each vertex a label uniformly at random from $[k]$, then independently add edges with probability $(1 + \frac{k-1}{k}\varepsilon)\frac{d}{n}$ between vertices with the same label and with probability $(1 - \frac{\varepsilon}{k})\frac{d}{n}$ between vertices with different labels.

Note that, when the bias parameter $\varepsilon = 0$, the model reduces to Erdős-Rényi random graphs with average degree $d$, denoted by $\mathbb{G}(n, \frac{d}{n})$.

Given a graph sampled from SSBM($n, \frac{d}{n}, \varepsilon, k$), the most fundamental problem is to recover the hidden community labels of the vertices, or equivalently to recover the *community membership matrix* $M^\circ$ given by

$$M^\circ_{i,j} := \mathbf{1}_{x^\circ_i = x^\circ_j} - \frac{1}{k} \quad (i, j \in [n]),$$

where $x^\circ_i \in [k]$ is the label of the $i$-th vertex. In this work, we consider *weak recovery* of $M^\circ$, that is to find a matrix $M$ which *correlates* with $M^\circ$ in the following sense.

**Definition 1.2** (Recovery rate and weak recovery in the SBM)**.** For any $\delta \in [0, 1]$, an algorithm achieves recovery rate $\delta$ in the $k$-stochastic block model, if given a random graph sampled from SSBM($n, \frac{d}{n}, \varepsilon, k$), it outputs a nonzero matrix $M \in \mathbb{R}^{n \times n}$ such that with constant probability,

$$\langle M, M^\circ \rangle \geqslant \delta \|M\|_{\mathrm{F}} \|M^\circ\|_{\mathrm{F}}.$$

If the recovery rate $\delta$ satisfies $\delta \geqslant \Omega(1)$, then the algorithm is said to achieve *weak recovery*.

The difficulty of achieving weak recovery in the SBM appears to be closely related to the choice of parameters $\varepsilon, d, k$. In particular, [Mas14, MS15] give polynomial-time algorithms for weak recovery above the KS threshold $\varepsilon^2 d \geqslant k^2$. On the other hand, while it is known that *exponential-time* algorithms can achieve weak recovery below this threshold (when $k \geqslant 5$) [BMNN16], it is widely believed that no polynomial-time algorithms exist that achieve weak recovery when $\varepsilon^2 d < k^2$.

**Rigorous evidence for average case complexity.**    To provide rigorous evidence for the innate hardness of weak recovery below the Kesten-Stigum threshold, one could follow two general approaches. The first is to construct a reduction from problems widely believed to be hard (such as *planted clique* [BBH19, BB20] or *learning with errors (LWE)* [BRST20, GVV22, Tie24]). However, this approach is unlikely to be successful for our problem as no such reductions are known for (other) average-case problems with constant sharp statistical threshold. The second approach is to prove *unconditional* lower bounds that rule out certain classes of algorithms. For example, those based on sums of squares [BHK$^+$19, KMOW17, JPR$^+$22,

---

[1]For ease of notation, we will use the adjacency matrix and the graph interchangeably.

MRX20], statistical queries [BKW03, Fel17, BBH$^+$20], or low-degree polynomials [HKP$^+$17, HS17, Hop18, KWB19]. As it appears that significant technical barriers have to be overcome to prove lower bounds against the former two classes for average-case problems with sharp statistical threshold, we focus on the latter.

**The low-degree method for hypothesis testing.** In recent years, the low-degree method has emerged as a standard tool for providing rigorous evidence for computational hardness in average-case problems [Hop18, KWB19]. Inspired by the fact that thresholding the likelihood ratio function provides optimal algorithms for hypothesis testing, the low-degree method provides a heuristic for average-case computational hardness by proving lower bounds against the low-degree projection of the likelihood ratio between two distributions from the hypothesis class. In fact, previous works[Hop18, BBK$^+$21a] has provided low-degree hardness evidence for the related hypothesis testing problems on distinguishing the stochastic block model and Erdős-Rényi distribution with probability $1 - o(1)$. However, significant barrier needs to be overcome for extending their computational lower bound to weak recovery below KS threshold. The reason is that we want to rule out recovery algorithms which only need to succeed with constant probability, while below the Kesten-Stigum threshold, the two distributions considered in [Hop18, BBK$^+$21a] can be distinguished with probability strictly larger than 1/2 by counting triangles in the graph [BM17]. [2]

**Low-degree recovery lower bound.** In this paper, we focus on the implications of the following conjecture of [MW23] in the context of the SBM[3].

**Conjecture 1.3** (Low-degree conjecture). *Let P be a distribution from the k-stochastic block model and Q be a distribution of Erdős-Rényi random graphs. For (randomized) functions $f : \mathbb{R}^{n \times n} \to \mathbb{R}$, consider the parameter*

$$R_{P,Q}(f) := \frac{\mathbb{E}_{Y \sim P} f(Y)}{\sqrt{\mathbb{E}_{Y \sim Q}(f(Y))^2}}. \tag{1.1}$$

*Suppose that for any arbirary constant $\delta \in (0, 1]$, and every polynomial $f(\cdot)$ of degree at most $n^\delta$, we have $R_{P,Q}(f) \leqslant O(1)$. Then, for any function $f(\cdot)$ computable in time $\exp(n^{0.99\delta})$ taking values in $[0, 1]$ satisfying $\mathbb{E}_{Y \sim P} f(Y) \geqslant \Omega(1)$ , we have $R_{P,Q}(f) \leqslant O(1)$.*

The parameter $R_{P,Q}(\cdot)$ in (1.1) is motivated by Le Cam's method: if the maximum of $R_{P,Q}(f)$ over all computable functions $f$ is bounded by $O(1)$, no algorithm can distinguish between the distributions $P$ and $Q$ with high probability. Intuitively, Conjecture 1.3 tells us that, if the maximum of $R_{P,Q}(f)$ is bounded over all low-degree polynomials, it is in fact bounded over all efficiently computable functions.

Assuming this conjecture for $P$ given by the symmetric SBM and $Q$ given by the Erdős-Rényi graph distribution, we aim to clarify the relation between the upper bounds for the *low-degree likelihood ratio* (proved in [Hop18, BBK$^+$21a]) and lower bounds for computationally efficient algorithms for the SBM. Specifically, we address the following question:

**Question 1.4.** Assuming Conjecture 1.3, can we rule out polynomial-time algorithms achieving weak recovery (or even non-trivial error rate) in the stochastic block model below the Kesten-Stigum threshold, when the number of communities is a universal constant?

**Implications for learning stochastic block model.** A potential computational-statistical gap similar to weak recovery can also be observed in the closely related problem of learning the parameters of the stochastic block model [Xu17]. Although [LG24] established a low-degree recovery lower bound for learning the probability matrix in the symmetric SBM, their result does not imply a lower bound for learning the parameters $d, \varepsilon$, as their hard

---

[2]Similar detection-recovery gap in success probabilities is also recently revealed by [HS24] in the context of planted cliques.

[3]The original conjecture is stated for the closely related spiked Wigner model.

instance is given by a symmetric SBM with fixed parameters. Moreover, when $k \lesssim \log(n)$ (rather than constant), their lower bound cannot rule out polynomial-time algorithms for achieving the minimax error rate.

As such, we address the following question in this paper:

**Question 1.5.** Assuming Conjecture 1.3, can we provide rigorous evidence for a computational-statistical gap in the error rate of learning the parameters of the stochastic block model?

## 2 Main result

### 2.1 Computational lower bound for weak recovery in stochastic block model

We provide the first rigorous evidence that no polynomial-time algorithms can achieve recovery rate $n^{-0.49}$ with constant probability below the Kesten-Stigum threshold, assuming Conjecture 1.3.

**Theorem 2.1** (Computational lower bound below the KS threshold, see Theorem B.1 for the full statement)**.** *Let $k, d \in \mathbb{N}^+$ be such that $k \leqslant O(1), d \leqslant O(1)$. Assume that for any $d' \in \mathbb{N}^+$ such that $0.999d \leqslant d' \leqslant d$, Conjecture 1.3 holds with distribution P given by $SSBM(n, \frac{d'}{n}, \varepsilon, k)$ and distribution Q given by the Erdős-Rényi graph model $\mathbb{G}(n, \frac{d'}{n})$. Then, no $\exp(n^{0.99})$ time algorithm can achieve recovery rate $n^{-0.49}$ in the k-stochastic block model when $\varepsilon^2 d \leqslant 0.99k^2$.*

Note that the algorithm which outputs a matrix $\hat{M}$ reflecting a random partition of the vertices into $k$ communities only achieves vanishing recovery rate $\delta \lesssim 1/\sqrt{n}$. In contrast, recall that above the KS threshold (when $\varepsilon^2 d/k^2 > 1$), polynomial-time algorithms can achieve a recovery rate in $\Omega(1)$ (i.e., weak recovery). To our knowledge, this is the first result showing such a sharp transition in the recovery rate that can be achieved by efficient algorithms above and below the KS threshold[4].

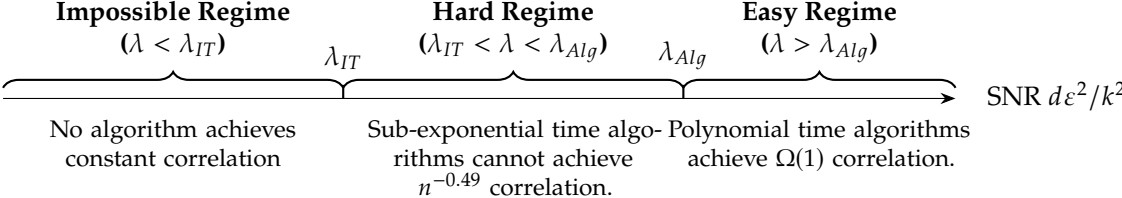

Under a strengthened low-degree conjecture (see Conjecture A.2 below), our lower bound extends to the regime where the number of communities can be as large as $n^{o(1)}$.

**Theorem 2.2** (Computational lower bound for diverging number of blocks)**.** *Let $k, d \in \mathbb{N}^+$ be such that $k \leqslant n^{o(1)}, d \leqslant n^{o(1)}$. Assume that for any $d' \in \mathbb{N}^+$ such that $0.999d \leqslant d' \leqslant d$, Conjecture A.2 holds with distribution P given by $SSBM(n, \frac{d'}{n}, \varepsilon, k)$ and distribution Q given by the Erdős-Rényi graph model $\mathbb{G}(n, \frac{d'}{n})$. Then no $\exp(n^{0.99})$ time algorithm can achieve recovery rate $n^{-0.49}$ in the k-stochastic block model when $\varepsilon^2 d \leqslant 0.99k^2$.*

**Concurrent work.** A concurrent work [SW25] also provides rigorous evidence for computational lower bound below KS threshold based on low-degree polynomials. They give the first unconditional low-degree recovery lower bound below the KS threshold in the stochastic block model. In our setting, for the task of weak recovery (which is to say, achieving constant recovery rate), their lower bound directly rules out estimators based on degree $n^\delta$

---

[4]For small number of communities $k = 2, 3, 4$, there is no information-computation gap, and thus no hard regime. In these scenarios, we suspect that there is sharp phase transition of recovery rate from $n^{-0.49}$ to constant information-theoretically.

polynomials, which captures many natural candidates of algorithms. Such a lower bound is beyond reach with techniques from [Hop18, BBK+21a]. For this, they introduce significantly new techniques in analyzing the low-degree polynomials.

In comparison, we give evidence that sub-exponential time algorithms cannot achieve recovery rate $n^{-0.49}$ with constant probability below the KS threshold, while polynomial time algorithms are known to achieve weak recovery above the threshold. Our techniques are based on relating the rate of recovery to the low-degree conjecture formulated in [MW23]. Notably, we did not prove new low-degree lower bounds for our main results, but exploited the existing results from [Hop18, BBK+21a].

In a concurrent work, [Li25] established computational lower bounds for random graph matching below sharp thresholds, also based on a strengthened version of the low-degree conjecture. However, their polynomial-time reduction between hypothesis testing with lopsided success probability and recovery differs significantly from ours.

As we discuss next, our result also has implications for other learning tasks in the stochastic block model.

## 2.2 Computational lower bound for learning stochastic block model

We give computational lower bounds for learning the stochastic block model under two different error metrics: learning the edge connection probability matrix and the block graphon function.

**Definition 2.3** (Edge connection probability matrix for the SBM). In the symmetric stochastic block model $\text{SSBM}(n, \frac{d}{n}, \varepsilon, k)$, the edge connection probability matrix $\theta^\circ \in [0,1]^{n \times n}$ has entries $\theta^\circ_{i,j} = (1 + \frac{(k-1)\varepsilon}{k})\frac{d}{n}$ if $i, j$ belong to the same community and $\theta^\circ_{i,j} = (1 - \frac{\varepsilon}{k})\frac{d}{n}$ if $i, j$ belong to different communities.

Given a random graph sampled from distribution $\text{SSBM}(\varepsilon, d, k, n)$, the simple polynomial-time algorithm based on $k$-SVD outputs a matrix $\theta \in [0,1]^{n \times n}$ such that $\mathbb{E}\|\theta - \theta^\circ\|_F^2 \leq O(2k \cdot d)$ [Xu17, LG24]. On the other hand, exponential-time algorithms based on maximum-likelihood can give an estimator which achieves the optimal error rate $\mathbb{E}\|\theta - \theta^\circ\|_F^2 \leq O(\log(k) \cdot d + k^2)$. We give rigorous evidence for the hardness of learning the edge connection probability matrix of symmetric SBMs, by proving the following computational lower bound.

**Theorem 2.4** (Computational lower bound for learning the SBM). *Let $k, d \in \mathbb{N}^+$ be such that $k \leq n^{o(1)}, d \leq o(n)$. Assume that for any $d' \in \mathbb{N}^+$ such that $0.999d \leq d' \leq d$, Conjecture A.2 holds with distribution $P$ given by $\text{SSBM}(n, \frac{d'}{n}, \varepsilon, k)$ and distribution $Q$ given by Erdős-Rényi graph model $\mathbb{G}(n, \frac{d}{n})$. Then given graph $G \sim \text{SSBM}(n, \frac{d}{n}, \varepsilon, k)$, no $\exp(n^{0.99})$ time algorithm can output $\theta \in [0,1]^{n \times n}$ achieving error rate $\|\theta - \theta^\circ\|_F^2 \leq 0.99kd/4$ with constant probability, where $\theta^\circ$ is the sampled edge connection probability matrix.*

Our computational lower bound matches the guarantees of known efficient algorithms in [Xu17] up to constant factors.[5] In comparison, [LG24] show that degree-$\ell$ polynomials cannot give error rate better than $O(kd/\ell^4)$. For standard low-degree conjectures [Hop18, KWB19, SW22], to give evidence of hardness for polynomial-time algorithms, the polynomial degree $\ell$ needs to be taken as large as $\log(n)$. Therefore, their lower bound on the error rate can only match the guarantees of existing algorithm up to logarithmic factors. As a result, they cannot give evidence of a computational-statistical gap for the error rate when $k = O(\log(n))$.[6]

Another error metric considered in [KTM+15, BCS15, BCSZ18, CDD+24] is learning the graphon function. In the context of the symmetric SBM, the graphon function is block-wise constant and given by:

---

[5]For completeness, we state this algorithmic result in Appendix F.2.

[6]We also obtain an unconditional low-degree recovery lower bound for learning the $k$-stochastic block model when $k \leq n^{0.001}$ in Appendix D.

**Definition 2.5** (Graphon in the symmetric SBM). Let $d, k \in \mathbb{N}^+$ and $\varepsilon \in [0, 1]$. Consider the symmetric stochastic block model $\mathrm{SSBM}(n, \frac{d}{n}, \varepsilon, k)$. Let $B^\circ \in [0, 1]^{k \times k}$ be the community connection probability matrix with diagonal entries given by $(1 - \frac{\varepsilon(k-1)}{k})\frac{d}{n}$ and non-diagonal entries given by $(1 + \frac{\varepsilon}{k})\frac{d}{n}$. Let $\gamma : [0, 1] \to [k]$ be a mapping such that $\gamma(x) = \lceil kx \rceil$. Then a function $W^\circ : [0, 1] \times [0, 1] \to [0, 1]$ is a graphon generating distribution $\mathrm{SSBM}(n, \frac{d}{n}, \varepsilon, k)$ if $W^\circ(x, y) = B^\circ_{\gamma(x), \gamma(y)}$.

We note that in contrast with the edge connection probability matrix, the graphon function only depends on the parameters of the distribution $\varepsilon, d, k$. Previous works [BCS15, BCSZ18, KTM+15, CDD+24] consider the following distance metric between graphons:

**Definition 2.6** (Gromov-Wasserstein distance between graphons). Let functions $W_1, W_2 : [0, 1] \times [0, 1] \to [0, 1]$. We consider the Gromov-Wasserstein distance metric

$$\mathrm{GW}(W_1, W_2) := \sqrt{\min_\phi \int_0^1 \int_0^1 \left(W_1(\phi(x), \phi(y)) - W_2(x, y)\right)^2 dx\, dy}$$

where the minimum is taken over all measure-preserving bijective mappings.

Given a graph sampled from $\mathrm{SSBM}(n, \frac{d}{n}, \varepsilon, k)$, our goal is to output a graphon $\hat{W}$ minimizing $\mathrm{GW}(\hat{W}, W^\circ)$. [KTM+15] obtains the minimax error rate

$$\mathrm{GW}(\hat{W}, W^\circ) \lesssim \sqrt{\frac{d^2}{n^2} \cdot \left(\frac{k^2}{nd} + \frac{\log(k)}{d} + \sqrt{\frac{k}{n}}\right)}.$$

However, existing polynomial-time algorithms [Xu17, CDD+24] can only achieve error rate

$$\mathrm{GW}(\hat{W}, W^\circ) \lesssim \sqrt{\frac{d^2}{n^2} \cdot \left(\frac{k}{d} + \sqrt{\frac{k}{n}}\right)}.$$

Although there is a lower bound showing that the error term $\frac{d}{n}(k/n)^{1/4}$ is information-theoretically necessary[BCS15], it is not clear whether polynomial time algorithms can achieve better error rate than $\frac{d}{n}\sqrt{k/d}$, especially if the graph is sparse.

In this paper, we give the first rigorous evidence for a computational-statistical gap in learning the graphon function when the number of blocks $k$ is a sufficiently large constant.

**Theorem 2.7** (Computational lower bound for learning block graphon function). *Let $k, d \in \mathbb{N}^+$ be such that $k \leqslant O(1), d \leqslant o(n)$. Assume that Conjecture 1.3 holds with distribution $P$ given by $\mathrm{SSBM}(n, \frac{d}{n}, \varepsilon, k)$ and distribution $Q$ given by Erdős-Rényi graph model $\mathbb{G}(n, \frac{d}{n})$. Then no $\exp(n^{0.99})$ time algorithm can output a $\mathrm{poly}(n)$-block graphon function $\hat{W} : [0, 1] \times [0, 1] \to [0, 1]$ such that $\mathrm{GW}(\hat{W}, W^\circ) \leqslant \frac{d}{3n}\sqrt{\frac{k}{d}}$ with $1 - o(1)$ probability under distribution $P$ and distribution $Q$.*

In comparison, [LG24] do not provide any lower bound for learning the graphon function since their hard instance is a symmetric SBM with fixed distribution parameters $\varepsilon, d, k$.

## 3 Techniques

### 3.1 Lower bounds for weak recovery

In this section, we give an overview of the techniques we use to prove lower bounds for weak recovery in the stochastic block model with constant number of blocks $k$ and constant average degree $d$. That is, an overview of our proof of (a special case of) Theorem 2.1.

Suppose for a contradiction that we have a polynomial-time recovery algorithm for $\mathrm{SSBM}(n, \frac{d}{n}, \varepsilon, k)$, $\varepsilon^2 d \leqslant 0.99k^2$, that achieves recovery rate $\delta \geqslant \Omega(n^{-0.49})$, in the sense

of Definition 1.2. Using this algorithm, we will construct a function $f(\cdot)$, which can be evaluated in polynomial time, such that $R_{P,Q}(f) \geq n^{\Omega(1)}$. Here, $R_{P,Q}(\cdot)$ is the parameter (1.1) for the distributions $P = \mathrm{SSBM}(n, \frac{d}{n}, \varepsilon, k)$ and $Q = \mathbb{G}(d, n)$. Assuming the low-degree conjecture (Conjecture 1.3) for this $P$ and $Q$, this implies that there exists a low-degree polynomial $f'$ with $R_{P,Q}(f') \geq \omega(1)$. But, since $\varepsilon^2 d \leq 0.99 k^2$ is below the KS threshold, this leads to a contradiction with the low-degree lower bound of [Hop18] (Theorem A.1).

In the remainder of this section, we show how to construct the function $f(\cdot)$, and give a sketch of the proof that $R_{P,Q}(f) \geq n^{\Omega(1)}$.

**Regularization via correlation-preserving projection.** We begin with the following tool, which allows us to regularize the estimators of the community membership matrix $M^\circ$ provided by the recovery algorithm. Suppose that $\hat{M}_0$ is a matrix achieving correlation $\delta$ with $M^\circ$, i.e., satisfying $\langle \hat{M}_0, M^\circ \rangle \geq \delta \|\hat{M}_0\|_{\mathrm{F}} \|M^\circ\|_{\mathrm{F}}$. We show that $\hat{M}_0$ can be projected into a (small) convex set $\mathcal{K} \subseteq \mathbb{R}^{n \times n}$ containing $M^\circ$, while preserving the correlation (up to a constant). Concretely, the convex set $\mathcal{K}$ here is given by

$$\mathcal{K} := \left\{ M \in [-1/\delta, 1/\delta]^{n \times n} : M + \frac{1}{k\delta} \mathbf{1}\mathbf{1}^\top \succeq 0, \mathrm{Tr}(M + \frac{1}{k\delta} \mathbf{1}\mathbf{1}^\top) \leq n/\delta \right\}. \tag{3.1}$$

In particular, elements of $\mathcal{K}$ have bounded entries and bounded nuclear norm, which will be crucial in later steps of the proof where we apply a Bernstein inequality. To achieve this, we make use of the *correlation preserving projection* from [HS17] (see Theorem F.1), which projects $\hat{M}_0$ onto a matrix $\hat{M} \in \mathcal{K}$ satisfying

$$\langle \hat{M}, M^\circ \rangle \geq \Omega(1) \cdot \delta \|\hat{M}\|_{\mathrm{F}} \|M^\circ\|_{\mathrm{F}},$$

In addition, Theorem F.1 promises that $\|\hat{M}\|_{\mathrm{F}} = \Theta(\|M^\circ\|_{\mathrm{F}})$. Thus, we find that

$$\langle \hat{M}, M^\circ \rangle \geq \Omega(1) \cdot \delta \|\hat{M}\|_{\mathrm{F}} \|M^\circ\|_{\mathrm{F}} \geq \Omega(1) \cdot \delta \|M^\circ\|_{\mathrm{F}}^2 \geq \delta \cdot \Omega(n^2). \tag{3.2}$$

Importantly, the correlation preserving projection can be implemented in polynomial time via semidefinite programming. See Lemma B.5 for details.

**Testing statistics via cross validation.** The basic idea is to construct $f(\cdot)$ via cross validation. Given a random graph $G$, we construct a subgraph $G_1$ with the same vertex set by subsampling each edge in $G$ independently with probability $1 - \eta$, where $\eta > 0$ is a small constant. Note that if $G$ is drawn from $\mathrm{SSBM}(n, \frac{d}{n}, \varepsilon, k)$, then $G_1$ is distributed according to $\mathrm{SSBM}(n, (1-\eta)\frac{d}{n}, \varepsilon, k)$. If $G$ is drawn from $\mathbb{G}(n, \frac{d}{n})$, then $G_1$ is distributed according to $\mathbb{G}(n, (1-\eta)\frac{d}{n})$. We run the polynomial-time recovery algorithm on $G_1$ to obtain an estimate $\hat{M} \in \mathbb{R}^{n \times n}$ of the community membership matrix $M^\circ$, which we regularize using the correlation preserving projection discussed above. Let $Y_2$ denote the adjacency matrix of $G_2 := G \setminus E(G_1)$. Our function $f(\cdot)$ is then defined as

$$f(Y) := \begin{cases} 1, & \text{if } \langle \hat{M}, Y_2 - \frac{\eta d}{n} \mathbf{1}\mathbf{1}^\top \rangle \geq n^{0.51}, \\ 0, & \text{otherwise.} \end{cases} \tag{3.3}$$

See Algorithm 3.1 for an overview of the construction of $f(\cdot)$. It remains to show that $R_{P,Q}(f) \geq n^{\Omega(1)}$. For this, we establish a *lower bound* on the expectation $\mathbb{E}_{Y \sim P} f(Y)$ of $f$ under graphs drawn from the SBM and an *upper bound* on the expectation $\mathbb{E}_{Y \sim Q} f(Y)$ of $f$ under Erdős-Rényi random graphs.

> **Algorithm 3.1** (Test function $f(\cdot)$ used in the proof of Theorem 2.1).
> **Input:** A graph $G$ with $n$ vertices, given by its adjacency matrix $Y$.
> **Output:** Test function $f(Y) \in \{0, 1\}$.
> **Algorithm:**
>
> 1. Obtain a subgraph $G_1$ of $G$ by subsampling each edge with probability $1 - \eta$.
>
> 2. Obtain an estimator $\hat{M}_0$ by running a recovery algorithm on the graph $G_1$.
>
> 3. Obtain $\hat{M}$ by projecting $\hat{M}_0$ onto the set $\mathcal{K}$ defined in (3.1) using the correlation preserving projection.
>
> 4. Return $f(Y) = \mathbb{1}\{\langle \hat{M}, Y_2 - \frac{\eta d}{n} \mathbb{1}\mathbb{1}^\top \rangle \geqslant n^{0.51}\}$, where $Y_2$ is the adjacency matrix of $G \setminus E(G_1)$.

**Lower bound on the expectation under the SBM..**    First, we give a lower bound on the expectation of $f$ under the distribution $\text{SSBM}(n, \frac{d}{n}, \varepsilon, k)$, i.e, on

$$\mathop{\mathbb{E}}_{Y \sim P} f(Y) = \mathop{\mathbb{P}}_{Y \sim P}\left[ \langle \hat{M}, Y_2 - \frac{\eta d}{n} \mathbb{1}\mathbb{1}^\top \rangle \geqslant n^{0.51} \right].$$

To do so, note that we may decompose $Y_2 - \frac{\eta d}{n} \mathbb{1}\mathbb{1}^\top = \frac{\varepsilon \eta d}{n} M^\circ + W_2$, where $W_2$ is a random matrix whose entries are independent with mean zero. Then, we have

$$\langle Y_2 - \frac{\eta d}{n} \mathbb{1}\mathbb{1}^\top, \hat{M} \rangle = \frac{\varepsilon \eta d}{n} \langle M^\circ, \hat{M} \rangle + \langle W_2, \hat{M} \rangle.$$

The first term on the RHS above is large with constant probability by (3.2). We would like to apply a Bernstein inequality to the second term, but the matrices $W_2$ and $\hat{M}$ are not independent. However, as we show in Lemma E.3, they are *approximately independent* in the sense that there exists a zero-mean symmetric matrix $\tilde{W}_2$ with independent entries, independent of $\hat{M}$, so that each entry in $\tilde{W}_2 - W_2$ has variance bounded by $O(d^2/n^2)$.

For ease of presentation, we ignore the difference between $\tilde{W}_2$ and $W_2$ for now, and assume that $W_2$ and $\hat{M}$ are independent. In this case, we note that $\langle W_2, \hat{M} \rangle$ can be written as the summation of independent zero-mean random variables, namely

$$\langle W_2, \hat{M} \rangle = \sum_{i,j} W_2(i, j)\hat{M}(i, j),$$

where $W_2(i, j)\hat{M}(i, j) \leqslant O(1/\delta)$ for each $i, j \in [n]$. (Here, we have used that $\hat{M} \in \mathcal{K}$). Moreover, since $\|\hat{M}\|_{\text{F}}^2 = \Theta(\|M^\circ\|_{\text{F}}^2) = \Theta(n^2)$, we have

$$\sum_{i,j} \hat{M}(i, j)^2 \, \mathbb{E}\big[W_2(i, j)^2\big] \lesssim \frac{d}{n} \sum_{i,j} \hat{M}(i, j)^2 \leqslant O(n^2 \cdot \frac{d}{n}) = O(nd).$$

By the Bernstein inequality, and using the fact that $\delta \geqslant \Omega(n^{-0.49})$, we then have

$$\mathbb{P}\left[ \sum_{i,j} W_2(i, j)\hat{M}(i, j) \geqslant n^{0.501} \right] \leqslant \exp\left(-n^{0.001}\right).$$

As result, when $d \leqslant O(1), k \leqslant O(1), \eta = \Theta(1)$, with constant probability, we have

$$\langle Y_2 - \frac{\eta d}{n}, \hat{M} \rangle \geqslant \frac{\eta \varepsilon d}{n} \langle M^\circ, \hat{M} \rangle - n^{0.501} \gtrsim \delta n - n^{0.501} \gtrsim n^{0.51} - n^{0.501} \geqslant \Omega(n^{0.51}).$$

Therefore, we have $\mathbb{E}_{Y \sim P} f(Y) \geqslant \Omega(1)$.

**Upper bound under the null distribution.**  Next, we give an upper bound on the expectation of $f$ under the Erdős-Rényi distribution $\mathbb{G}(n, \frac{d}{n})$. Our proof shares many ingredients with the proof of the lower bound in the previous section. We show that with high probability under the distribution $\mathbb{G}(d, n)$, we have

$$g(Y) := \left| \langle Y_2 - \frac{\eta d}{n} \mathbb{1}\mathbb{1}^\top, \hat{M} \rangle \right| = o(n^{0.51}).$$

To do so, we again apply the argument that $Y_2 - \frac{\eta d}{n} \mathbb{1}\mathbb{1}^\top$ and $\hat{M}$ are approximately independent. In particular, let $W_2 = Y_2 - \frac{\eta d}{n} \mathbb{1}\mathbb{1}^\top$, for some i.i.d. zero-mean symmetric matrix $\tilde{W}_2$, independent of $\hat{M}$, so that each entry in $\tilde{W}_2 - W_2$ has variance bounded by $d^2/n^2$. By the triangle inequality, we have

$$\left| \langle Y_2 - \frac{\eta d}{n} \mathbb{1}\mathbb{1}^\top, \hat{M} \rangle \right| \leqslant \left| \langle W_2 - \tilde{W}_2, \hat{M} \rangle \right| + \left| \langle \tilde{W}_2, \hat{M} \rangle \right|.$$

For simplicity, we again ignore the difference between $\tilde{W}_2$ and $W_2$ here. By the same reasoning as above, and again relying on the properties of $\hat{M}$ guaranteed by the correlation preserving projection, we have the Bernstein inequality

$$\mathbb{P}\left[ \left| \langle \tilde{W}_2, \hat{M} \rangle \right| \geqslant n^{0.501} \right] \leqslant \exp\left( -n^{0.01} \right).$$

As result, when $d \leqslant O(1)$ and $k \leqslant O(1)$, we have $g(Y) \leqslant o(n^{0.501})$ with probability at least $1 - \exp(-n^{0.01})$ and thus $\mathbb{E}_{Y \sim Q} f(Y) \leqslant \exp(-n^{0.01})$.

**Finishing the proof.**  Using the lower and upper bound established above, and the fact that $f(\cdot) \in \{0, 1\}$, we get that

$$\frac{\mathbb{E}_{Y \sim P} f(Y) - \mathbb{E}_{Y \sim Q} f(Y)}{\sqrt{\mathrm{Var}_{Y \sim Q}(f(Y))}} \geqslant \frac{\Omega(1)}{\exp(-n^{0.01})} \geqslant \exp(n^{0.01}) \geqslant \omega(1).$$

## 3.2   Lower bound for learning the stochastic block model

In this section, we give an overview of the techniques used to prove our results on learning the stochastic block model, stated in Section 2.2.

**Lower bound for learning edge connection probability matrix.**  We sketch the proof of Theorem 2.4. We show that if an $O(\exp(n^{0.99}))$-time algorithm can learn the edge connection probability matrix $\theta^\circ$ such that with constant probability, the error rate $\|\hat{\theta} - \theta^\circ\|_F^2 \leqslant 0.99kd$, then an algorithm with running time $\exp(n^{0.99})$ can achieve weak recovery when $\varepsilon^2 d \geqslant 0.99k^2$. The key observation is that, for the symmetric stochastic block model, the edge connection probability matrix is given by $\theta^\circ = \frac{(1-\eta)\varepsilon d}{n} M^\circ + \frac{(1-\eta)d}{n}$, where $M^\circ \in \{1 - 1/k, -1/k\}^{n \times n}$ is the community membership matrix. Therefore, when the estimation error is smaller than $0.99\sqrt{kd}$, the estimator $\hat{\theta} - \frac{d}{n}$ achieves weak recovery under the distribution $\mathrm{SSBM}(n, \frac{d}{n}, \varepsilon, k)$, which contradicts the extended low-degree conjecture (Conjecture A.2).

**Lower bound for learning graphon function.**  We sketch the proof of Theorem 2.7. Let $W_0$ be the graphon function underlying the distribution $\mathbb{G}(n, \frac{d}{n})$ and $W_1$ be the graphon function underlying the distribution $\mathrm{SSBM}(n, \frac{d}{n}, \varepsilon, k)$. We then have $\mathrm{GW}(W_0, W_1) \geqslant \frac{d}{n}\sqrt{\frac{0.99k}{d}}$ when $\varepsilon^2 d \geqslant 0.99k^2$.

Now suppose there is a polynomial-time algorithm which, given a random graph $G$ sampled from an arbitrary symmetric $k$-stochastic block model, outputs an $n$-block graphon function

$\hat{W} : [0, 1] \times [0, 1] \to [0, 1]$ achieving error $\frac{d}{3n}\sqrt{\frac{k}{d}}$ with probability $1 - o(1)$. Then one can construct a testing statistic by taking

$$f(Y) = \begin{cases} 1, & \text{if } \mathrm{GW}(\hat{W}, W_0) \leqslant \frac{3d}{n}\sqrt{\frac{k}{d}}, \\ 0, & \text{otherwise.} \end{cases}$$

We have $f(Y) = 1$ with probability $1 - o(1)$ under the distribution $\mathrm{SSBM}(n, \frac{d}{n}, \varepsilon, k)$ and $f(Y) = 0$ with probability $1 - o(1)$ under the distribution $\mathbb{G}(n, \frac{d}{n})$. Therefore, we have $R_{P,Q}(f) \geqslant \omega(1)$. Since the function $f(\cdot)$ can be evaluated in polynomial time, this contradicts the low-degree lower bound (Theorem A.1), assuming Conjecture 1.3.

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

# A  Preliminaries

## A.1  Low-degree framework

**Low-degree likelihood ratio lower bound in the SBM.**  The low-degree likelihood ratio lower bound is a standard framework to provide evidence of hardness for hypothesis testing problems. [Hop18, BBK$^+$21a] prove the following theorem[7] on the low-degree lower bound for the stochastic block model:

**Theorem A.1** (Low-degree lower bound for SBM, Thm. 2.20 in [BBK$^+$21a])**.**  *Let $d = o(n)$, $k = n^{o(1)}$ and $\varepsilon \in [0, 1]$. Let $\mu : \{0, 1\}^{n \times n} \to \mathbb{R}$ be the relative density of $SSBM(n, d, \varepsilon, k)$ with respect to $G\left(n, \frac{d}{n}\right)$. Let $\mu^{\leq \ell}$ be the projection of $\mu$ to the degree-$\ell$ polynomials with respect to the norm induced by $G\left(n, \frac{d}{n}\right)$ For any constant $\delta > 0$,*

$$\left\| \mu^{\leq \ell} \right\| \text{ is } \begin{cases} \geq n^{\Omega(1)}, & \text{if } \varepsilon^2 d > (1 + \delta)k^2, \quad \ell \geq O(\log n) \\ \leq O_\delta \left(\exp(k^2)\right), & \text{if } \varepsilon^2 d < (1 - \delta)k^2, \quad \ell < n^{0.99} \end{cases}$$

Assuming the low-degree conjectures in [Hop18, KWB19], this gives rigorous hardness evidence for distinguishing

- planted distribution $P$: symmetric $k$-stochastic block model $SSBM(n, \frac{d}{n}, \varepsilon, k)$,
- null distribution $Q$: Erdős-Rényi random graph $\mathbb{G}(n, \frac{d}{n})$,

with probability $1 - o(1)$ when $k$ is a universal constant and $\varepsilon^2 d \leq 0.99 k^2$. However, even assuming the low-degree conjectures for hypothesis testing from [Hop18, KWB19], these works do not rule out polynomial-time weak recovery algorithms under our definition (i.e., algorithms that achieve constant correlation with constant probability).

**Extended low-degree hypothesis.**  To show our lower bounds in the regime where $k$ is polylogarithmic, Conjecture 1.3 is not sufficient. Instead, we rely on the following stronger low-degree hypothesis from [MW23].

**Conjecture A.2** (Extended low-degree conjecture)**.**  *Let $P$ be a distribution from the $k$-stochastic block model and $Q$ be a distribution of Erdős-Rényi random graphs. Consider the hypothesis testing problem between $Y \sim P$ and $Y \sim Q$ for distribution $P$ and $Q$. Let $R_{P,Q}(f) := \frac{\mathbb{E}_{Y \sim P} f(Y)}{\sqrt{\mathbb{E}_{Y \sim Q}(f(Y))^2}}$. Let $\delta \in (0, 1]$. For any (randomized) function $f(\cdot)$ computable in time $\exp(n^{0.99\delta})$ taking values in $[0, 1]$ and satisfying $\mathbb{E}_P f(Y) \geq \Omega(1)$, we have*

$$R_{P,Q}(f) \lesssim \max_{deg(f) \leq n^\delta} R_{P,Q}(f).$$

When $\max_{\deg(f) \leq n^\delta} R_{P,Q}(f) \leq O(1)$, the conjecture is reduced to Conjecture 1.3. The extended low-degree hypothesis is closely related to the low-degree lower bound for random optimization problems (see [GJW24]).

## A.2  Organization

The rest of the paper is organized as follows. We present our main proof ideas in Section 3. In Appendix B, we give the formal proof of our computational lower bounds for recovery algorithms conditional on the low-degree conjectures (i.e., the proof of Theorem 2.1). In

---

[7]Although the original theorem statement is for constant $k, d$, it is easy to see in their analysis that the lower bound holds when $d = o(n)$. Also a weaker version of the theorem is stated in Thm. 8.6.1 of [Hop18].

Appendix C, we give proofs of our computational lower bounds for parameter learning algorithms conditional on the low-degree conjectures (i.e., the proof of Theorem 2.4 and Theorem 2.7). In Appendix E, we introduce some facts from probability theory used in our paper. In Appendix F, we introduce some existing algorithms from the literature that are used in our paper. In Appendix G, we clarify the upper bound on the low-degree likelihood ratio when the number of blocks $k$ diverges, which is implicitly obtained in [BBK$^+$21a].

## B  Computational lower bound for recovery

In this section, we prove Theorem 2.1 by showing that there exists an efficient algorithm that reduces testing to weak recovery in SBM. We will show that there exists a efficiently computable testing function (shown in Algorithm B.2) that is large with constant probability if the input is sampled from $\text{SSBM}(n, \frac{d}{n}, \varepsilon, k)$ and is small with high probability if the input is sampled from $\mathbb{G}(n, d/n)$. This will lead to a contradiction with low-degree lower bounds of testing if we assume Conjecture 1.3.

Before describing the algorithm, we restate Theorem 2.1 here for completeness.

**Theorem B.1** (Full version of Theorem 2.1). *Let $k, d \in \mathbb{N}^+$ be such that $k \leqslant O(1), d \leqslant n^{o(1)}$. Assume that for any $d' \in \mathbb{N}^+$ such that $0.999d \leqslant d' \leqslant d$, Conjecture 1.3 holds for distribution $P = \text{SSBM}(n, \frac{d'}{n}, \varepsilon, k)$ and distribution $Q = \mathbb{G}(n, \frac{d'}{n})$. Then for any small constants $\delta_1, \delta_2$, no $\exp(n^{0.99})$ time algorithm can achieve recovery rate $n^{-0.5+\delta_1}$ in the $k$-stochastic block model when $\varepsilon^2 d \leqslant (1 - \delta_2)k^2$.*

The reduction that we consider is the following.

---

**Algorithm B.2** (Reduction from testing to weak recovery).
**Input:** A random graph $G$ with equal probability sampled from Erdős-Rényi model or stochastic block model, and target recovery rate $\delta$, parameters $\varepsilon, k, d$.
**Output:** Testing statistics $g(Y) \in \mathbb{R}$, where $Y$ is the adjacency matrix.
**Algorithm:**

1. Let $\eta = 0.001\delta_2$, where $\delta_2 = 1 - \varepsilon^2 d/k^2$. Obtain subgraph $G_1$ by subsampling each edge with probability $1 - \eta$, and let $G_2 = G \setminus G_1$.

2. Obtain estimator $\hat{M}_0$ by running weak recovery algorithm on graph $G_1$.

3. Obtain $\hat{M}$ by applying correlation preserving projection (see Theorem F.1) on $\hat{M}_0$ to the set $\mathcal{K} = \left\{ M \in [-1/\delta, 1/\delta]^{n \times n} : M + \frac{1}{k\delta}\mathbf{1}\mathbf{1}^\top \geq 0, \text{Tr}(M + \frac{1}{k\delta}\mathbf{1}\mathbf{1}^\top) \leqslant n/\delta \right\}$.

4. Return testing statistics $g(Y) = \langle \hat{M}, Y_2 - \frac{\eta d}{n}\mathbb{1}\mathbb{1}^\top \rangle$, where $Y_2$ is the adjacency matrix for the graph $G_2$.

---

To prove Theorem B.1, we will show that the testing statistics $g(Y)$ from Algorithm B.2 satisfies the following two lemmas.

**Lemma B.3.** *Let $Y$ be the adjacency matrix of the graph sampled from the symmetric $k$-stochastic block model $\text{SSBM}(n, \frac{d}{n}, \varepsilon, k)$ and $M^\circ \in \{-1/k, 1 - 1/k\}^{n \times n}$ be the corresponding community membership matrix. Suppose that $\langle \hat{M}_0, M^\circ \rangle \geqslant n^{-0.5+\delta_1}\|\hat{M}_0\|_\text{F}\|M^\circ\|_\text{F}$ and $\|\hat{M}\|_\text{F} = \Theta(\|M^\circ\|_\text{F})$. Then Algorithm B.2 outputs testing statistics $g(Y) \in \mathbb{R}$ such that $g(Y) \geqslant \Omega\left(n^{0.5(1+\delta_1)}\right)$.*

**Lemma B.4.** *Let $Y$ be the adjacency matrix of the graph sampled from Erdős-Rényi random graph $\mathbb{G}(n, d/n)$. With probability at least $1 - \exp(-n^{0.001\delta_1})$, Algorithm B.2 outputs $g(Y) \leqslant O(n^{0.5+\delta_1/3})$ in polynomial time.*

Combining Lemma B.3 and Lemma B.4, Theorem 2.1 follows as a corollary.

*Proof of Theorem 2.1.* Suppose that there is a $\exp(n^{0.99})$ time algorithm which outputs estimator $\hat{M}_0$ such that $\langle \hat{M}_0, M^\circ \rangle \geq n^{-0.5+\delta_1} \|\hat{M}_0\|_{\mathrm{F}} \|M^\circ\|_{\mathrm{F}}$. Let $f(Y) = \mathbf{1}_{g(Y) \geq 0.001 n^{0.5+\delta_1/2}}$. When $\varepsilon^2 d \geq \Omega(k^2)$, combining Lemma B.3 and Lemma B.4, we have

$$\frac{\mathbb{E}_P \, f(Y)}{\sqrt{\mathrm{Var}_Q(f(Y))}} \geq \exp(n^{0.001\delta_1}) \,.$$

By the low-degree likelihood ratio upper bound Theorem A.1, when $\varepsilon^2 d \leq (1-\delta_2)k^2$, we have

$$\max_{\deg(f) \leq n^{0.01}} \frac{\mathbb{E}\, f(Y)}{\sqrt{\mathrm{Var}_Q(f(Y))}} \leq \exp(k^2) \,.$$

Since $f(Y)$ can be evaluated in $O(\exp(n^{0.99}))$ time, assuming Conjecture 1.3, we then have

$$\frac{\mathbb{E}\, f(Y)}{\sqrt{\mathrm{Var}_Q(f(Y))}} \lesssim \max_{\deg(f) \leq n^{0.01}} \frac{\mathbb{E}\, f(Y)}{\sqrt{\mathrm{Var}_Q(f(Y))}} \leq O(1) \,,$$

which leads to a contradiction. As a result, assuming Conjecture 1.3, we cannot achieve weak recovery in $\exp(n^{0.99})$ time when $\varepsilon^2 d \leq (1-\delta_2)k^2$. $\qquad\square$

## B.1 Correlation preserving projection

In this part, we prove that we can project the estimator into the set of matrices with bounded entries and bounded nuclear norm, while preserving correlation.

**Lemma B.5.** *Let $M^\circ \in \{-1/k, 1-1/k\}^{n \times n}$ be a symmetric matrix with rank-$(k+1)$. For any $\delta \leq O(1)$, given matrix $\hat{M}_0$ such that $\langle \hat{M}_0, M^\circ \rangle \geq \delta \|\hat{M}_0\|_{\mathrm{F}} \|M^\circ\|_{\mathrm{F}}$, there is a polynomial time algorithm which outputs $\hat{M} \in \mathcal{K}$ such that $\langle \hat{M}, M^\circ \rangle \geq \Omega(1) \cdot \delta \|\hat{M}\|_{\mathrm{F}} \|M^\circ\|_{\mathrm{F}}$ and $\|\hat{M}\|_{\mathrm{F}} \geq \Omega(\|M^\circ\|_{\mathrm{F}})$, where*

$$\mathcal{K} = \left\{ M \in [-1/\delta, 1/\delta]^{n \times n} : M + \frac{1}{k\delta} \mathbf{1}\mathbf{1}^\top \succeq 0 \,, \mathrm{Tr}(M + \frac{1}{k\delta} \mathbf{1}\mathbf{1}^\top) \leq n/\delta \right\} .$$

*Proof.* We apply the correlation preserving projection from [HS17] (restated in Theorem F.1). By definition, $M^\circ = X^\circ(X^\circ)^\top - \frac{1}{k} \mathbf{1}\mathbf{1}^\top$ is in $\mathcal{K}$. Let $N$ be the matrix that minimizes $\|N\|_{\mathrm{F}}$ subject to $N \in \mathcal{K}'$ and $\langle N, \hat{M}_0 \rangle \geq \delta \|M^\circ\|_{\mathrm{F}} \|\hat{M}_0\|_{\mathrm{F}}$, where

$$\mathcal{K}' = \left\{ M \in [-1, 1]^{n \times n} : M + \frac{1}{k} \mathbf{1}\mathbf{1}^\top \succeq 0 \,, \mathrm{Tr}(M + \frac{1}{k} \mathbf{1}\mathbf{1}^\top) \leq n \right\} .$$

Using ellipsoid method, this semidefinite program can be solved in polynomial time. By Theorem F.1, we have $\langle N, M^\circ \rangle \geq \Omega(1) \cdot \delta \|N\|_{\mathrm{F}} \|M^\circ\|_{\mathrm{F}}$ and $\|N\|_{\mathrm{F}} \geq \delta \|M^\circ\|_{\mathrm{F}}$. We let $\hat{M} = \frac{\|M^\circ\|_{\mathrm{F}}}{\|N\|_{\mathrm{F}}} \cdot N$. Then it follows that $\hat{M} \in \mathcal{K}$, $\|\hat{M}\|_{\mathrm{F}} = \|M^\circ\|_{\mathrm{F}}$ and $\langle \hat{M}, M^\circ \rangle \geq \Omega(\delta) \|\hat{M}\|_{\mathrm{F}} \cdot \|M^\circ\|_{\mathrm{F}}$. $\quad\square$

## B.2 Proof of Lemma B.3

In this section, we prove Lemma B.3.

*Proof of Lemma B.3.* We consider the decomposition that $Y_2 - \frac{\eta d}{n} \mathbf{1}\mathbf{1}^\top = \frac{\varepsilon \eta d}{n} M^\circ + W_2$ where $W_2$ is a symmetric random matrix with independent and zero mean entries. By Lemma E.3, there exists an i.i.d zero mean symmetric matrix $\tilde{W}_2$ that is independent with $Y_1$, and satisfies that the entries in $\tilde{W}_2 - W_2$ are independent with zero mean and have variance bounded by $O(d^3/n^3)$, conditioning on the subsampled graph $Y_1$ and community matrix $M^\circ$. As result, we have

$$\left\langle Y_2 - \frac{\eta d}{n} \mathbf{1}\mathbf{1}^\top, \hat{M} \right\rangle = \left\langle \frac{\varepsilon \eta d}{n} M^\circ, \hat{M} \right\rangle + \langle W_2 - \tilde{W}_2, \hat{M} \rangle + \langle \tilde{W}_2, \hat{M} \rangle \,.$$

For the first term $\langle \frac{\varepsilon \eta d}{n} M^\circ, \hat{M} \rangle$, it follows from [Lemma B.5](#) that

$$\langle M^\circ, \hat{M} \rangle \geqslant \Omega\left(\frac{\varepsilon \eta d}{n}\right) \delta \|\hat{M}\|_F \|M^\circ\|_F$$

$$\geqslant \Omega\left(\frac{\delta \varepsilon \eta d}{n}\right) \|M^\circ\|_F^2.$$

As with probability at least $1 - \exp(-n^{0.001})$, we have $\|M^\circ\|_F^2 \geqslant \Omega(n^2)$, and as result $\langle M^\circ, \hat{M} \rangle \geqslant \Omega(n\delta \varepsilon d)$.

For bounding the second term $\langle W_2 - \tilde{W}_2, \hat{M} \rangle$, we condition on the subsampled graph $Y_1$ and the community matrix $M^\circ$. With probability at least $1 - \exp(-n^{\delta_1})$, we have

$$|\langle W_2 - \tilde{W}_2, \hat{M} \rangle| \leqslant \|W_2 - \tilde{W}_2\|_F \cdot \|\hat{M}\|_F \lesssim \sqrt{\frac{d^3}{n^3} \cdot n^{2+\delta_1} \cdot n^2} = \sqrt{n^{1+\delta_1} d^3}.$$

For the third term $\langle \tilde{W}_2, \hat{M} \rangle$, we again conditional on the subsampled graph $Y_1$ and the community matrix $M^\circ$. We note that it can be written as the summation of independent zero-mean random variables

$$\langle \tilde{W}_2, \hat{M} \rangle = \sum_{i,j} \tilde{W}_2(i,j) \hat{M}(i,j).$$

where $\tilde{W}_2(i,j) \hat{M}(i,j)$ are independent zero mean variables bounded by $O(1/\delta)$ for all $i \leqslant j$. Moreover, we have

$$\sum_{i,j} \hat{M}(i,j)^2 \mathbb{E}\left[\tilde{W}_2(i,j)^2\right] \lesssim \frac{d}{n} \sum_{i,j} \hat{M}(i,j)^2 \leqslant O(n^2 \cdot \frac{d}{n}) = O(nd).$$

By Bernstein inequality, we have

$$\mathbb{P}\left[\left|\sum_{i,j} \tilde{W}_2(i,j) \hat{M}(i,j)\right| \geqslant 100t\right] \leqslant \exp\left(-t^2/(nd + t/\delta)\right).$$

Taking $t = n^{(1+\delta_1)/2} \sqrt{d}$ and $\delta \geqslant n^{-0.5+\delta_1}$, we have

$$\mathbb{P}\left[\left|\sum_{i,j} \tilde{W}_2(i,j) \hat{M}(i,j)\right| \geqslant n^{0.5(1+\delta_1)} \sqrt{d}\right] \leqslant \exp\left(-n^{\delta_1/2}\right).$$

As a result, when $d \leqslant n^{o(1)}, k \leqslant n^{o(1)}, \delta \geqslant n^{-0.5+\delta_1}, \varepsilon = \Theta(1/\sqrt{d})$, with constant probability, we have

$$\langle Y_2 - \frac{\eta d}{n} \mathbb{1}\mathbb{1}^\top, \hat{M} \rangle \geqslant \Omega(n^{0.5+\delta_1} \sqrt{d}).$$

$\square$

## B.3 Proof of [Lemma B.4](#)

In this section, we prove [Lemma B.4](#).

*Proof of [Lemma B.4](#).* We will use the fact that $Y_2 - \frac{\eta d}{n} \mathbb{1}\mathbb{1}^\top$ and $\hat{M}$ are approximately independent. More precisely, let $W_2 = Y_2 - \frac{\eta d}{n} \mathbb{1}\mathbb{1}^\top$, by [Lemma E.3](#), there exists symmetric zero mean matrix $\tilde{W}_2$ with independent entries such that each entry in $\tilde{W}_2 - W_2$ has zero mean variance bounded by $O(d^3/n^3)$ conditioning on $\hat{M}$. By triangle inequality, we have

$$g(Y) = \left|\langle Y_2 - \frac{\eta d}{n} \mathbb{1}\mathbb{1}^\top, \hat{M} \rangle\right| \leqslant \left|\langle W_2 - \tilde{W}_2, \hat{M} \rangle\right| + \left|\langle \tilde{W}_2, \hat{M} \rangle\right|.$$

For bounding the first term $\langle W_2 - \tilde{W}_2, \hat{M} \rangle$, we condition on the subsampled graph $Y_1$. With probability at least $1 - \exp(-n^{\delta_1/3})$, we have

$$|\langle W_2 - \tilde{W}_2, \hat{M} \rangle| \leq \|W_2 - \tilde{W}_2\|_F \cdot \|\hat{M}\|_F \lesssim \sqrt{\frac{d^3}{n^3} \cdot n^{2+\delta_1/3} \cdot n^2} = \sqrt{d^3 n^{1+\delta_1/3}}.$$

For the second term, we note that $\langle \tilde{W}_2, \hat{M} \rangle$ can be written as the summation of independent zero-mean random variables

$$\langle \tilde{W}_2, \hat{M} \rangle = \sum_{i,j} \tilde{W}_2(i,j) \hat{M}(i,j).$$

where $\tilde{W}_2(i,j)\hat{M}(i,j)$ are independent zero mean variables bounded by $O(1/\delta)$ for $i \leq j$. Moreover, we have

$$\sum_{i,j} \hat{M}(i,j)^2 \, \mathbb{E}\big[\tilde{W}_2(i,j)^2\big] \lesssim \frac{d}{n} \sum_{i,j} \hat{M}(i,j)^2 \leq O(n^2 \cdot \frac{d}{n}) = O(nd).$$

By Bernstein inequality, we have

$$\mathbb{P}\left[\left|\sum_{i,j} \tilde{W}_2(i,j)\hat{M}(i,j)\right| \geq 100t\right] \geq \exp\left(-t^2/(nd + t/\delta)\right).$$

Taking $t = n^{0.5+\delta_1/3}\sqrt{d}$ and $\delta \geq n^{-0.5+\delta_1}$, we have

$$\mathbb{P}\left[\left|\sum_{i,j} \tilde{W}_2(i,j)\hat{M}(i,j)\right| \geq n^{0.5+\delta_1/3}\right] \leq \exp\left(-n^{0.001\delta_1}\right).$$

$\square$

### B.4 Proof of Theorem 2.2

In this part, we give the proof of Theorem 2.2, which is the same as the proof of Theorem 2.1 except that we assume stronger low-degree conjecture.

*Proof of Theorem 2.2.* Suppose that there is a polynomial time algorithm which outputs estimator $\hat{M}_0$ such that $\langle \hat{M}_0, M^\circ \rangle \geq n^{-0.5+\delta_1}\|\hat{M}_0\|_F\|M^\circ\|_F$. Let $f(Y) = \mathbf{1}_{g(Y) \geq 0.001 n^{0.5+\delta_1/2}}$. When $0.001 k^2 \leq \varepsilon^2 d \leq (1-\delta_2)k^2$, combining Lemma B.3 and Lemma B.4, we have

$$\frac{\mathbb{E}_P f(Y)}{\sqrt{\mathrm{Var}_Q(f(Y))}} \geq \exp(n^{0.001}).$$

Since $f(Y)$ can be evaluated in $O(\exp(n^{0.001}))$ time, assuming Conjecture 1.3, by [Hop18](stated in Theorem A.1), we have

$$\frac{\mathbb{E} f(Y)}{\sqrt{\mathrm{Var}_Q(f(Y))}} \lesssim \max_{\deg(f) \leq n^{0.99}} \frac{\mathbb{E} f(Y)}{\sqrt{\mathrm{Var}_Q(f(Y))}} \leq \exp(k^2).$$

When $k^2 \leq n^{0.001}$, this leads to a contradiction. As a result, assuming Conjecture A.2, we cannot achieve recovery rate $n^{-0.5+\delta_1}$ in polynomial time when $\varepsilon^2 d \leq (1 - \Omega(1))k^2$. $\square$

## C   Computational lower bound for learning stochastic block model

### C.1   Computational lower bound for learning the edge connection probability matrix

In this section, we prove Theorem 2.4 by showing that there exists an efficient algorithm that reduces testing to learning in SBM. The reduction of algorithm Algorithm C.2 is similar to

that of Algorithm B.2. The proof of Theorem 2.4 is also a similar proof by contradiction to the proof of Theorem 2.1.

Before describing the algorithm, we restate Theorem 2.4 here for completeness.

**Theorem C.1** (Restatement of Theorem 2.4). *Let $k, d \in \mathbb{N}^+$ be such that $k \leqslant n^{o(1)}, d \leqslant o(n)$. Assume that for any $d' \in \mathbb{N}^+$ such that $0.999d \leqslant d' \leqslant d$, Conjecture A.2 holds with distribution P given by $SSBM(n, \frac{d'}{n}, \varepsilon, k)$ and distribution Q given by Erdős-Rényi graph model $\mathbb{G}(n, \frac{d'}{n})$. Then given graph $G \sim SSBM(n, \frac{d}{n}, \varepsilon, k)$, no $\exp(n^{0.99})$ time algorithm can output $\theta \in [0,1]^{n \times n}$ achieving error rate $\|\theta - \theta^\circ\|_F^2 \leqslant 0.99kd/4$ with constant probability, where $\theta^\circ$ is the ground truth sampled edge connection probability matrix.*

The reduction that we consider is the following.

---

**Algorithm C.2** (Reduction from testing to learning).
**Input:** A random graph $G$ with equal probability sampled from Erdős-Rényi model or stochastic block model.
**Output:** Testing statistics $g(Y) \in \mathbb{R}$, where $Y$ is the centered adjacency matrix
**Algorithm:**

1. Obtain subgraph $G_1$ by subsampling each edge with probability $1 - \eta = 0.999$, and let $G_2 = G \setminus G_1$.

2. Run learning algorithm on $G_1$, and obtain estimator $\hat{\theta} \in \mathbb{R}^{n \times n}$

3. Obtain $\hat{M}$ by running correlation preserving projection on $\hat{\theta} - \frac{d}{n}\mathbf{1}\mathbf{1}^\top$ to the set $\mathcal{K} = \{M \in [-1,1]^{n \times n} : M + \frac{1}{k}\mathbf{1}\mathbf{1}^\top \succeq 0, \text{Tr}(M + \frac{1}{k}\mathbf{1}\mathbf{1}^\top) \leqslant n\}$.

4. Construct the testing statistics $g(Y) = \langle \hat{M}, Y_2 - \frac{\eta d}{n}\mathbf{1}\mathbf{1}^\top \rangle$, where $Y_2$ is the adjacency matrix for the graph $G_2$.

---

Before proving Theorem 2.4, we first show the relationship between learning edge connection probability and weak recovery.

**Lemma C.3.** *Consider the distribution of $SSBM(n, \frac{d}{n}, \varepsilon, k)$ with $d \leqslant n^{o(1)}$. Suppose give graph $Y \sim SSBM(n, \frac{d}{n}, \varepsilon, k)$, the estimator $\hat{\theta} \in \mathbb{R}^{n \times n}$ achieves error rate $\|\hat{\theta} - \theta^\circ\|_F \leqslant \frac{1}{2}\sqrt{0.99kd}$ with constant probability, then $\hat{\theta} - d/n$ achieves weak recovery when $\varepsilon^2 d \geqslant 0.99k^2$.*

*Proof.* By the relation between edge connection probability matrix $\theta^\circ$ and the community matrix $M^\circ$, We have

$$\langle \hat{\theta} - \frac{d}{n}\mathbf{1}\mathbf{1}^\top, M^\circ \rangle = \langle \hat{\theta} - \theta^\circ, M^\circ \rangle + \langle \theta^\circ - \frac{d}{n}\mathbf{1}\mathbf{1}^\top, M^\circ \rangle = \langle \hat{\theta} - \theta^\circ, M^\circ \rangle + \langle \frac{\varepsilon d}{n}M^\circ, M^\circ \rangle.$$

For the first term, since with constant probability, $\|\hat{\theta} - \theta^\circ\|_F \leqslant \sqrt{0.99kd}$, we have

$$\left| \langle \hat{\theta} - \theta^\circ, M^\circ \rangle \right| \leqslant \|M^\circ\|_F \|\hat{\theta} - \theta^\circ\|_F \leqslant \|M^\circ\|_F \sqrt{0.99kd}.$$

For the second term, since with overwhelming high probability, $\|M^\circ\|_F \geqslant \frac{n}{\sqrt{k}}(1 - \frac{1}{k})$, we have

$$\langle \frac{\varepsilon d}{n}M^\circ, M^\circ \rangle = \frac{\varepsilon d}{n}\|M^\circ\|_F^2 \geqslant \frac{\varepsilon d}{2\sqrt{k}}\|M^\circ\|_F.$$

Therefore, when $\varepsilon^2 d > 0.999k^2$, we have

$$\langle \hat{\theta} - \frac{d}{n}\mathbf{1}\mathbf{1}^\top, M^\circ \rangle \geqslant \frac{\varepsilon d}{2\sqrt{k}}\|M^\circ\|_F - \|M^\circ\|_F \frac{\sqrt{0.99kd}}{2} \geqslant \Omega\left( \frac{\varepsilon d\|M^\circ\|_F}{\sqrt{k}} \right).$$

On the other hand, by triangle inequality

$$\left\| \hat{\theta} - \frac{d}{n}\mathbf{1}\mathbf{1}^\top \right\|_F \leqslant \|\hat{\theta} - \theta^\circ\|_F + \left\| \theta^\circ - \frac{d}{n}\mathbf{1}\mathbf{1}^\top \right\|_F \leqslant O(\sqrt{kd} + \frac{\varepsilon d}{\sqrt{k}}) \leqslant O\left( \varepsilon d/\sqrt{k} \right),$$

Therefore we have

$$\langle \hat{\theta} - \frac{d}{n} \mathbf{1}\mathbf{1}^\top, M^\circ \rangle \geq \Omega(\|M^\circ\|_F \cdot \|\hat{\theta} - \frac{d}{n} \mathbf{1}\mathbf{1}^\top\|_F).$$

We thus conclude that with constant probability, $\hat{\theta} - \frac{d}{n} \mathbf{1}\mathbf{1}^\top$ achieves weak recovery when $\varepsilon^2 d \geq 0.99 k^2$. $\qquad\square$

With Lemma C.3, the proof of lower bound for learning the edge connection probability matrix of stochastic block model follows as a corollary.

*Proof of Theorem 2.4.* By Lemma C.3, suppose an $\exp(n^{0.99})$ time algorithm achieves error rate less than $0.99\sqrt{kd}$ in estimating the edge connection probability matrix, then in Algorithm C.2, $\hat{\theta} - \frac{d}{n}$ achieves weak recovery when $\varepsilon^2 d = 0.99 k^2$. We let $f(Y) = \mathbf{1}_{g(Y) \geq 0.001 \varepsilon^2 d^2 / k}$.

We show that with constant probability under $P$, we have $f(Y) = 1$. We essentially follow the proof of Lemma B.3 with $\delta$ taken as a constant, except that we take a different strategy for bounding $\langle W_2 - \tilde{W}_2, \hat{M} \rangle$. By Lemma E.2, we have, with probability at least $1 - o(1)$, the following spectral radius bounds on the symmetric random matrices

$$\|W_2 - \tilde{W}_2\|_{op} \leq O\left(\sqrt{d \log(n)} \cdot \sqrt{\frac{d}{n}}\right).$$

Therefore, by Trace inequality, we have

$$
\begin{aligned}
|\langle W_2 - \tilde{W}_2, \hat{M} \rangle| &= |\langle W_2 - \tilde{W}_2, \hat{M} + \frac{1}{k\delta} \mathbf{1}\mathbf{1}^\top \rangle - \langle W_2 - \tilde{W}_2, \frac{1}{k\delta} \mathbf{1}\mathbf{1}^\top \rangle| \\
&\leq |\langle W_2 - \tilde{W}_2, \hat{M} + \frac{1}{k\delta} \mathbf{1}\mathbf{1}^\top \rangle| + |\langle W_2 - \tilde{W}_2, \frac{1}{k\delta} \mathbf{1}\mathbf{1}^\top \rangle| \\
&\leq \|W_2 - \tilde{W}_2\|_{op} \operatorname{Tr}(\hat{M} + \frac{1}{k\delta} \mathbf{1}\mathbf{1}^\top) + \|W_2 - \tilde{W}_2\|_{op} \operatorname{Tr}(\frac{1}{k\delta} \mathbf{1}\mathbf{1}^\top) \\
&\leq O\left(\sqrt{d \log(n)} \cdot \sqrt{\frac{d}{n}}(1 + \frac{1}{k})\frac{n}{\delta}\right) \\
&= O\left((d + \frac{d}{k})\frac{\sqrt{n \log(n)}}{\delta}\right).
\end{aligned}
$$

With the same reasoning, by Lemma B.4, with probability at least $1 - \exp(-n^{0.001})$ under distribution $Q$, we have $f(Y) = 0$. Therefore, we have $R_{P,Q}(f) \geq \exp(n^{0.001})$. Since $f(A)$ can be evaluated in $O(\exp(n^{0.99}))$ time, assuming conjecture 1.3 we have

$$R_{P,Q}(f) := \frac{\mathbb{E} f(A)}{\sqrt{\operatorname{Var}_Q(f(A))}} \lesssim \max_{\deg(f) \leq n^{0.99}} \frac{\mathbb{E} f(A)}{\sqrt{\operatorname{Var}_Q(f(A))}}.$$

On the other hand, by low-degree lower bound stated in Theorem A.1, we have

$$\max_{\deg(f) \leq n^{0.99}} \frac{\mathbb{E} f(A)}{\sqrt{\operatorname{Var}_Q(f(A))}} \leq \exp(k^2).$$

Since we have $\exp(n^{0.001}) \gg \exp(k^2)$ when $k \leq n^{o(1)}$, this leads to a contradiction. $\qquad\square$

## C.2 Computational lower bound for learning graphon

In this part, we give formal proof of Theorem 2.7.

**Theorem C.4** (Restatement of [Theorem 2.7]). *Let $k, d \in \mathbb{N}^+$ be such that $k \leqslant O(1), d \leqslant o(n)$. Assume that Conjecture [1.3] holds with distribution $P$ given by $SSBM(n, \frac{d}{n}, \varepsilon, k)$ and distribution $Q$ given by Erdős-Rényi graph model $\mathbb{G}(n, \frac{d}{n})$. Then no $\exp(n^{0.99})$ time algorithm can output a poly$(n)$-block graphon function $\hat{W} : [0,1] \times [0,1] \to [0,1]$ such that $GW(\hat{W}, W^\circ) \leqslant \frac{d}{3n}\sqrt{\frac{k}{d}}$ with $1 - o(1)$ probability under distribution $P$ and distribution $Q$(where $W^\circ$ is the underlying graphon of the corresponding distribution).*

*Proof.* Let $W_0$ be the graphon function underlying the distribution $\mathbb{G}(n, \frac{d}{n})$ and $W_1$ be the graphon function underlying the distribution $SSBM(n, \frac{d}{n}, \varepsilon, k)$, we have $GW(W_0, W_1) \geqslant \frac{d}{n}\sqrt{\frac{0.99k}{d}}$ when $\varepsilon^2 d \geqslant 0.99k^2$.

Now suppose there is a polynomial time algorithm, which given random graph $G$ sampled from an arbitrary symmetric $k$-stochastic block model, outputs an $n$-block graphon function $\hat{W} : [0,1] \times [0,1] \to [0,1]$ achieving error $\frac{d}{3n}\sqrt{\frac{k}{d}}$ with probability $1 - o(1)$. Then one can construct the testing statistics by taking

$$f(Y) = \begin{cases} 1, & \text{if } GW(\hat{W}, W_0) \leqslant \frac{d}{3n}\sqrt{\frac{k}{d}} \\ 0, & \text{otherwise} \end{cases}$$

We have $f(Y) = 1$ with probability $1 - o(1)$ under the distribution of symmetric stochastic block model $SSBM(n, \frac{d}{n}, \varepsilon, k)$. By triangle inequality, we have $f(Y) = 0$ with probability $1 - o(1)$ under the distribution $\mathbb{G}(n, \frac{d}{n})$. Therefore we have $R_{P,Q}(f) \geqslant \omega(1)$.

Now since the function $\hat{W}$ can be represented as a symmetric matrix with poly$(n)$ number of rows and columns, and moreove since $W_0$ is a constant function,

$$GW(\hat{W}, W_0) = \int_0^1 \int_0^1 (\hat{W}(x, y) - W_0(x, y))^2 dx dy .$$

Therefore, the function $f(\cdot)$ can be evaluated in polynomial time. This contradicts the low-degree lower bound ([Theorem A.1]) assuming [Conjecture 1.3]. $\square$

## D   Low-degree recovery lower bound for learning dense stochastic block model

In this part, we give unconditional lower bound against low-degree polynomial estimators for the edge connection probability matrix in stochastic block model, via implementing reduction from hypothesis testing to weak recovery using low-degree polynomials. For simplicity, we focus on the dense graph.

**Theorem D.1** (Low-degree lower bound for learning). *Let $n \in \mathbb{N}^+$ and $\ell \leqslant n^{0.001}$. Let $d = \Theta(n)$. Let $\mathcal{F}_{n,\ell}$ be the set of degree-$\ell$ polynomials mapping from $n \times n$ symmetric matrices to $n \times n$ symmetric matrices. Suppose $\theta^\circ \in [0,1]^{n \times n}, Y \in \{0,1\}$ are edge connection probability matrix and adjacency matrix sampled from symmetric stochastic block model $SSBM(n, \frac{d}{n}, \varepsilon, k)$. Then for $k \leqslant n^{0.001}$, we have*

$$\min_{f \in \mathcal{F}_{n,\ell}} \max_{\varepsilon \in [0,1]} \mathop{\mathbb{E}}_{(Y, \theta^\circ) \sim SSBM(n, \frac{d}{n}, \varepsilon, k)} \|f(Y) - \theta^\circ\|_F^2 \geqslant \Omega(k \cdot n) .$$

### D.1   Construction of the low-degree polynomial

For simplicity, we define the community matrix of symmetric stochastic block model.

**Definition D.2** (Community matrix for stochastic block model). Under symmetric stochastic block model $SSBM(n, \frac{d}{n}, \varepsilon, k)$, we define the community matrix $X^\circ \in \{\pm 1\}$ as following: $X^\circ(i, j) = 1$ if vertex $i, j$ have the same community label and $X^\circ(i, j) = 0$ otherwise.

Given the polynomial function $f : \mathbb{R}^{n \times n} \to \mathbb{R}^{n \times n}$. We consider a graph with $2n$ nodes and randomly partition the nodes into two equal-sized sets $S_1$ and $S_2$. Let $X = \frac{n}{\varepsilon d}\left(f(Y_1) - \frac{d}{n}\right)$ where $Y_1$ is the subgraph induced by vertices in $S_1$. We construct the polynomial function $g : \mathbb{R}^{n \times n} \to \mathbb{R}$ as following:

$$g(Y) = \left\langle \left(Y_{12} - \frac{d}{n}\right)X\left(Y_{12} - \frac{d}{n}\right), Y_2 - \frac{d}{n}\right\rangle, \tag{D.1}$$

where $Y_{12} \in \mathbb{R}^{n \times n}$ is the adjacency matrix of the bipartite graph between vertices in $S_1$ and $S_2$, and $Y_2$ is the adjacency matrix of the induced subgraph supported on $S_2$.

We show the lower bound of this polynomial under the symmetric stochastic block model, and the upper bound of this polynomial under the Erdős-Rényi graph model.

**Lemma D.3.** *Let $\theta^\circ, Y$ be the edge connection probability matrix and adjacency matrix sampled from the planted distribution $SSBM(n, \frac{d}{n}, \varepsilon, k)$. Let $X_1^\circ$ be the community matrix of the subgraph induced by vertices in $S_1$. Suppose in Eq. (D.1), $\mathbb{E}\|X - X^\circ\|_F^2 \leqslant o(n^2)$, then we have $\mathbb{E}\, g(Y) \geqslant \left(\frac{\varepsilon d}{n}\right)^3 n^4$.*

**Lemma D.4.** *When the graph is sampled from the null distribution $\mathbb{G}(n, \frac{d}{n})$, we have $\mathbb{E}\, g(Y) = 0$ and $\sqrt{Var(g(Y))} \leqslant d^{3/2} \cdot n^{1-\Omega(1)}$.*

Combining Lemma D.3 and Lemma D.4, Theorem D.1 follows as a corollary:

*Proof of Theorem D.1.* Suppose there is a degree-$n^{0.001}$ polynomial $f : \mathbb{R}^{n \times n} \to \mathbb{R}^{n \times n}$ which gives error rate $o(n \cdot k)$. Let $X = \frac{n}{\varepsilon d}\left(f(Y) - \frac{d}{n}\right)$. Then we have

$$\|X - X^\circ\|_F = \frac{n^2}{\varepsilon^2 d^2}\|f(Y) - \theta^\circ\|_F^2 \leqslant o\left(\frac{n^2}{\varepsilon^2 d^2}kn\right) \leqslant o\left(\frac{k}{\varepsilon^2 d} \cdot \frac{n}{d} \cdot n^2\right).$$

When $\varepsilon^2 d \geqslant 0.001 k^2$ and $d = \Theta(n)$, we have $\mathbb{E}\|X - X^\circ\|_F^2 \leqslant o(n^2)$. combining Lemma D.3 and Lemma D.4, we have

$$\frac{\mathbb{E}\, g(Y)}{\sqrt{Var(g(Y))}} \geqslant n^{0.001}.$$

Since $g(Y)$ is a degree-$\ell$ polynomial with $\ell \leqslant n^{0.01}$, by [Hop18], we have

$$\frac{\mathbb{E}\, g(Y)}{\sqrt{Var(g(Y))}} \leqslant \exp(k^2).$$

When $\exp(k^2) \leqslant n^{0.001}$, this leads to a contradiction. As result, we conclude that no degree-$n^{0.001}$ polynomial can achieve error rate $o(nk)$. $\square$

## D.2 Proof of Lemma D.3

In this section, we analyze the property of the polynomial in Eq. (D.1) under the $k$-symmetric stochastic block model, and give a proof for Lemma D.3.

*Proof of Lemma D.3.* Let $X_{12}^\circ \in \{\pm 1\}^{n \times n}$ be the community matrix for the bipartite graph between $S_1$ and $S_2$, i.e for $i \in S_1$ and $j \in S_2$, we have $X_{12}^\circ(i, j) = 1$ if $i, j$ belongs to the same community and $X_{12}^\circ(i, j) = -1$ if $i, j$ belongs to different communities. Moreover, we let $X_1^\circ$ be the community matrix for the induced subgraph on $S_1$ and let $X_2^\circ$ be the community matrix for the induced subgraph on $S_2$. Then we have $Y_{12} = \frac{\varepsilon d}{n}X_{12}^\circ + W_{12}$, $Y_1 = \frac{\varepsilon d}{n}X_1^\circ + W_1$ and $Y_2 = \frac{\varepsilon d}{n}X_2^\circ + W_2$, where

- $W_{12}, W_1, W_2$ are independent,

- $(W_{12}, W_1, W_2)$ is independent with $(X_{12}^\circ, X_1^\circ, X_2^\circ)$,

- every entry in $W_{12}, W_1, W_2$ has zero mean.

Then we have

$$
\begin{aligned}
\mathbb{E}\, g(Y) &= \mathbb{E}\left\langle (Y_{12} - \frac{d}{n}) X (Y_{12} - \frac{d}{n}), Y_2 - \frac{d}{n} \right\rangle \\
&= \left(\frac{\varepsilon d}{n}\right)^3 \mathbb{E}\left\langle X_{12}^\circ X X_{12}^\circ, X_2^\circ \right\rangle + \mathbb{E}\left\langle W_{12} X W_{12}, W_2 \right\rangle + \frac{2\varepsilon d}{n} \mathbb{E}\left\langle W_{12} X X_{12}^\circ, W_2 \right\rangle \\
&= \left(\frac{\varepsilon d}{n}\right)^3 \mathbb{E}\left\langle X_{12}^\circ X X_{12}^\circ, X_2^\circ \right\rangle .
\end{aligned}
$$

Since $\mathbb{E}\left\langle X_{12}^\circ X_1^\circ X_{12}^\circ, X_2^\circ \right\rangle \geqslant \Omega(n^4)$ and

$$
\mathbb{E}\left\langle Y_{12}^\circ (X_1 - X_1^\circ) X_{12}^\circ, X_2^\circ \right\rangle \leqslant \sqrt{\mathbb{E}\|X_1 - X_1^\circ\|_F^2 \cdot \mathbb{E}\|X_{12}^\circ X_2^\circ X_{12}^\circ\|_F^2} \leqslant o(n^4) .
$$

Therefore, we have $\mathbb{E}\left\langle X_{12}^\circ X X_{12}^\circ, X_2^\circ \right\rangle \geqslant \Omega(n^4)$. and the claim follows. $\qquad\square$

### D.3  Proof of Lemma D.4

In this section, we analyze the property of the polynomial defined in Eq. (D.1), under the Erdős-Rényi graph distribution, and give a proof for Lemma D.4.

*Proof of Lemma D.4.* Under the Erdős-Rényi graph distribution, the entries in $Y_2 - \frac{d}{n}$ are i.i.d zero mean random variables, independent with $Y_{12}$ and $Y_2$(i.e the rest of the graph). As result, we have $\mathbb{E}\, g(Y) = 0$.

It remains to bound the variance of the polynomial under the Erdős-Rényi graph distribution, which is to say, we bound

$$
\mathbb{E}\, g(Y)^2 = \mathbb{E}\left\langle \left(Y_{12} - \frac{d}{n}\right) X \left(Y_{12} - \frac{d}{n}\right), Y_2 - \frac{d}{n} \right\rangle^2 .
$$

Let $W_{12} = Y_{12} - \frac{d}{n}$ and $W_2 = Y_2 - \frac{d}{n}$. The main observation is that $X, W_{12}, W_2$ are all independent. As result, let $Z = W_{12} X W_{12}$, we have

$$
\mathbb{E}\left\langle W_{12} X W_{12}, W_2 \right\rangle^2 = \sum_{ij} (\mathbb{E}\, W_2(i,j) Z(i,j))^2 = \sum_{ij} \mathbb{E}\, W_2^2(i,j)\, \mathbb{E}\, Z(i,j)^2 = \frac{d}{n} \mathbb{E}\|Z\|_F^2 .
$$

As $\|X\|_F \leqslant O(n)$ without loss of generality, and $\|W_{12}\| \leqslant \sqrt{d}\log(n)$ with overwhelming high probability, we have

$$
\mathbb{E}\|Z\|_F^2 \leqslant O\left(n^2 d^2 \log^4(n)\right) .
$$

Therefore we have

$$
\mathbb{E}\left\langle W_{12} X W_{12}, W_2 \right\rangle^2 \leqslant O\left(n d^3 \log^4(n)\right) .
$$

By taking the square root, we conclude the proof. $\qquad\square$

## E  Probability theory facts

In this section, we provide probability tools that we will need in the paper.

## E.1 Concentration of spectral radii of random matrices

The following concentration inequality for the spectral norm of the centered adjacency matrix of stochastic block model will be useful for our proofs.

**Theorem E.1** (Spectral norm bound for random matrices, theorem 2.7 in [BGBK20]). *Let $H \in \mathbb{R}^{n \times n}$ be a symmetric matrix whose upper triangular entries are independent zero mean random variables. Moreover, suppose that there exist $q > 0$ and $\kappa \geqslant 1$ such that*

$$\max_i \sum_j \mathbb{E}|H_{ij}|^2 \leqslant 1,$$

$$\max_{i,j} \mathbb{E}|H_{ij}|^2 \leqslant \kappa/n,$$

$$\max_{i,j}|H_{ij}| \leqslant 1/q.$$

*Then we have*

$$\mathbb{E}\|H\| \leqslant 2 + C\frac{\sqrt{\log(n)}}{q}.$$

*Moreover, we have*

$$\mathbb{P}[|\|H\| - \mathbb{E}\|H\|| \geqslant t] \leqslant 2\exp(-cq^2t^2).$$

As corollary, for stochastic block model, we have the following concentration inequality:

**Lemma E.2.** *Let $A$ be the adjacency matrix of a random graph with vertex $i, j$ independently connected with probability $\theta(i,j) \geqslant \Omega(1/n)$. Let $d = \frac{n-1}{n}\sum_{i,j}\theta(i,j)$ and suppose $\theta(i,j) \leqslant 2d/n$. Let $H = \frac{1}{\sqrt{2d}}(A - \theta)$. Then for every $t \geqslant 10000\log(n)$, for some small universal constant $c > 0$, we have*

$$\mathbb{P}[\|H\| \geqslant t] \leqslant 2\exp(-ct^2).$$

*Proof.* We verify that the matrix $H$ here satisfies the conditions in Theorem E.1. Crucially, since $\theta(i,j) \leqslant 2d/n$, we have $\mathbb{E}\,H_{i,j}^2 \leqslant 1/n$ First for each $i \in [n]$, we have $\sum_{j \in n}\mathbb{E}|H_{ij}|^2 \leqslant 1$. Finally, we have $\max_{i,j}|H_{i,j}| \leqslant 1$. Therefore by taking $\kappa = 1$ and $q = 1$ in Theorem E.1, we have $\mathbb{E}\|H\| \leqslant C\sqrt{\log(n)}$, and the concentration bound

$$\mathbb{P}[\|H\| \geqslant \mathbb{E}\|H\| + t] \leqslant 2\exp(-c't^2).$$

where $c'$ is a universal constant. Taking $t \geqslant 1000\log(n)$, we have the claim. $\qquad\square$

## E.2 Decoupling edge partition

In this section, we give a lemma that describes the approximate independence between edge sets of the subsampling process.

**Lemma E.3.** *Let $X \sim Ber(p)$, and let $X_1$ be obtained from $X$ by subsampling with probability $1-\eta$, i.e $X_1 = X\xi$, where $\xi \sim Ber(1-\eta)$ is independent of $X$. Let $X_2 = X - X_1$, and $\tilde{X}_2 = X_2 - \mathbb{E}[X_2|X_1] + \eta p$. Then we have $\mathbb{E}[\tilde{X}_2|X_1] = \mathbb{E}[X_2] = \eta p$ and $\mathbb{E}[(\tilde{X}_2 - X_2)^2|X_1] \leqslant O(p^3)$. Moreover, we have $|\tilde{X}_2 - X_2| \leqslant \eta p$.*

*Proof.* We first note that

$$\mathbb{E}[\tilde{X}_2|X_1] = \eta p = \mathbb{E}[X_2].$$

Next we note that

$$\mathbb{E}[X_2|X_1] = \frac{\eta p}{1 - (1-\eta)p}(1 - X_1),$$

Therefore, we have

$$\mathbb{E}[(\tilde{X}_2 - X_2)^2|X_1] = \mathbb{E}\left[\left(\eta p - \frac{\eta p(1 - X_1)}{(1 - (1-\eta)p)}\right)^2\right] \leqslant O(\eta^2 p^3)$$

$\qquad\square$

**Corollary E.4.** *Let $Y$ be the adjacency matrix of a random graph with each edge $(i, j)$ sampled with probability $p(i, j)$. Suppose that $p(i, j) \leqslant p$ for each $i, j \in [n]$. Let $Y_1$ be the adjacency matrix of the graph obtained by subsampling each edge in $Y$ with probability $1 - \eta$. Let $Y_2 = Y - Y_1$. Then there is a matrix $\tilde{Y}_2$ such that*

- *for every $t \geqslant \log(n)$, $\|\tilde{Y}_2 - Y_2\|_F^2 \leqslant t\eta p^3 n^2$ with probability at least $1 - \exp(-t)$,*

- *for every $i, j \in [n]$, $\mathbb{E}\,\tilde{Y}_2(i, j) = \mathbb{E}\,Y_2(i, j)$*

- *moreover $\tilde{Y}_2$ and $Y_1$ are independent,*

- *and finally the entries in $\tilde{Y}_2$ are independent.*

*Proof.* We construct the matrix $\tilde{Y}_2$ in the following way. For each $i, j$, we let $\tilde{Y}_2(i, j) = Y_2(i, j) - \mathbb{E}[Y_2(i, j)|Y_1] + \eta p(i, j)$. Then by [Lemma E.3](#), we have $\mathbb{E}\,\tilde{Y}_2(i, j) = \mathbb{E}\,Y_2(i, j)$ and $\mathbb{E}\big(\tilde{Y}_2(i, j) - Y_2(i, j)\big)^2 \leqslant O(\eta^2 p^3)$. In addition, we have $\big|\tilde{Y}_2(i, j) - Y_2(i, j)\big| \leqslant O(\eta p)$.

Furthermore $\tilde{Y}_2(i, j)$ and $Y_1$ are independent.

Finally since the upper triangular entries in $Y$ are independent, we have the upper triangular entries in $\tilde{Y}_2$ are independent. By Hoeffding bound, with probability at least $1 - \exp(-t)$, we have $\|\tilde{Y}_2 - Y_2\|_F^2 \leqslant t\eta p^3 n^2 \log(n)$. □

# F   Useful algorithmic results

In this section, we provide two algorithmic results from previous work that will be useful in our paper.

## F.1   Correlation preserving projection

Given a vector $P$ that has constant correlation with an unknown vector $Y$, [HS17] shows that one can project the vector $P$ into a convex set containing $Y$, and preserve the constant correlation with $Y$.

**Theorem F.1** (Correlation preserving projection, theorem 2.3 in [HS17]). *Let $\delta \in \mathbb{R}^+$ Let $C$ be a convex set and $Y \in C$. Let $P$ be a vector with $\langle P, Y \rangle \geqslant \delta \cdot \|P\| \cdot \|Y\|$. Then, if we let $Q$ be the vector that minimizes $\|Q\|$ subject to $Q \in C$ and $\langle P, Q \rangle \geqslant \delta \cdot \|P\| \cdot \|Y\|$, we have*

$$\langle Q, Y \rangle \geqslant \delta/2 \cdot \|Q\| \cdot \|Y\| . \tag{F.1}$$

*Furthermore, $Q$ satisfies $\|Q\| \geqslant \delta\|Y\|$.*

We include their proof here for completeness.

*Proof.* By construction, $Q$ is the Euclidean project of $0$ into the set $\{Q \in C | \langle P, Q \rangle \geqslant \delta\|P\|\|Y\|\}$. By Pythagorean inequality, the Euclidean projection into a set decreases distances to points into the set. Therefore, $\|Y - Q\|^2 \leqslant \|Y - 0\|^2$, which implies that $\langle Y, Q \rangle \geqslant \|Q\|^2/2$. Moreover, $\langle P, Q \rangle \geqslant \delta\|P\|\|Y\|$, which implies $\|Q\| \geqslant \delta\|Y\|$ by Cauchy-Schwartz. Thus, we can conclude that $\langle Y, Q \rangle \geqslant \delta/2 \cdot \|Y\| \cdot \|Q\|$. □

## F.2   Learning edge connection probability matrix via SVD

**Theorem F.2.** *When $d \geqslant \log(n)$, there is a polynomial time algorithm which given the adjacency matrix of a graph sampled from symmetric $k$-stochastic block model $SSBM(n, \frac{d}{n}, \varepsilon, k)$, returns an estimator $\hat{\theta} \in [0, 1]^{n \times n}$ such that $\|\theta^\circ - \theta\|_F^2 \leqslant kd$ with high probability.*

*Proof.* We take $\hat{\theta}$ as the best rank-$k$ approximation for the adjacency matrix. Then since $\|A - \theta^\circ\|_{\mathrm{op}} \leqslant \sqrt{kd}$ with high probability, we have $\|\hat{\theta} - A\|_{\mathrm{op}} \leqslant \sqrt{kd}$ with high probability. By triangle inequality, we have $\|\hat{\theta} - \theta^\circ\|_{\mathrm{op}} \leqslant 2\sqrt{d}$. As result, we have $\|\hat{\theta} - \theta^\circ\|_{\mathrm{F}} \leqslant 2\sqrt{kd}$. □

## G  Low-degree lower bound beyond constant number of blocks

By extending the result of [BBK+21b], we can get a more general bound (with respect to $k$) on low degree likelihood ratio of k-SBM. The proof of the extended result follows trivially from the proof of Theorem 2.20 of [BBK+21b]. Therefore, we only provide a proof sketch by pointing out the simple modifications that we need from the original proof.

**Theorem G.1** (Restatement of Theorem A.1). *Let $d = o(n), k = n^{o(1)}$ and $\varepsilon \in [0,1]$. Let $\mu : \{0,1\}^{n \times n} \to \mathbb{R}$ be the relative density of SBM $(n, d, \varepsilon, k)$ with respect to $G\left(n, \frac{d}{n}\right)$. Let $\mu^{\leqslant \ell}$ be the projection of $\mu$ to the degree-$\ell$ polynomials with respect to the norm induced by $G\left(n, \frac{d}{n}\right)$ For any constant $\delta > 0$,*

$$\left\|\mu^{\leqslant \ell}\right\| \text{ is } \begin{cases} \geqslant n^{\Omega(1)}, & \text{if } \varepsilon^2 d > (1+\delta)k^2, \quad \ell \geqslant O(\log n) \\ \leqslant O_\delta\left(\exp(k^2)\right), & \text{if } \varepsilon^2 d < (1-\delta)k^2, \quad \ell < n^{0.99} \end{cases}$$

*Proof.* In this proof, we stick to the notations of [BBK+21b]. The only modification we need is that the size of the $\delta$-net of the unit sphere in $\mathbb{R}^k$, denoted by $C(\delta, k)$, is equal to $\exp(O_\delta(k))$. The size of the $\delta$-net $C(\delta, k)$ is crucial in Proposition 6.4 and Proposition 6.5 of [BBK+21b] and is treated as constant in the proof of Theorem 2.16 and Theorem 2.20 of [BBK+21b].

By plugging $C(\delta, k) = O_\delta(\exp(k))$ into the upper bound of the small deviation term $L_1$ in the proof of Theorem 2.16 of [BBK+21b], it follows that we have $\left\|L^{\leqslant D}\right\| = O_\delta(\exp(k))$ for the likelihood ratio $L$ and $D \leqslant o(n/\log(n))$. Then, the bound on low-degree likelihood ratio of k-SBM follows from the same reduction as in proof of Theorem 2.20 of [BBK+21b], and we get $\varepsilon^2 d/(1 - d/n) \leqslant 1$. As $d = o(n)$, we get the claimed bound on low-degree likelihood ratio. □

## H  Conclusion and future directions

Based on low-degree heuristics, our paper gives rigorous evidence for a computational phase transition of recovery at the Kesten-Stigum threshold. We view our work as a first step in studying this phenomenon, leaving open many interesting questions:

- Below the Kesten-Stigum threshold, suppose we are given an initalization achieving recovery rate $n^{-0.49}$, could we boost the accuracy in polynomial time to get a weak recovery algorithm?

- Can we show a computational-statistical gap for learning the graphon function when the number of blocks satisfies $k \leqslant \sqrt{n}$ (as in [LG24])?

