# OpenReview forum: "Low-degree evidence for computational transition of recovery rate in stochastic block model"
_NeurIPS.cc/2025/Conference — NeurIPS 2025 spotlight_

### Official Review · Reviewer_owhx · 2025-06-16

**Clarity:** 3
**Significance:** 3
**Originality:** 4
**Rating:** 5
**Confidence:** 3

**Summary:**

This paper shows a reduction from a strong form of computational hardness of testing block models vs. ER null to strong computational hardness of recovery of block models.
The computational hardness is related to a conjecture of Moitra and Wein on low-degree hardness.

**Questions:**

0. It is very important to state the conjectured hardness that lead to the results of the paper. This is not conjecture 1.3 to be the best of my knowlege. It should be something like: There is no algorithm that runs in time 2^{O(n)} that can test between ER graphs and random graphs. Please be very explicit in the constant in the O here:
is it 2^{10 n} or 2^{0.9999 N}

1.Please discuss when are the intervals at the figure at page 4 empty.

2. Can you please provide an intuitive explanation for the advantage of cross validation

**Ethical Concerns:**

["NO or VERY MINOR ethics concerns only"]

**Final Justification:**

The paper makes nice progress in the low-degree hardness framework. The writing could be a little better, ideally better explaining the context and some of the main ideas of the proof.

**Limitations:**

The main limitation is the strong computational conjecture (that is never stated explicility ...) that is at the basis of this work.

**Quality:**

4

**Strengths And Weaknesses:**

The reduction is definitely interesting and a good contribution to the area.
The paper feels a bit rushed. I think the following could help the presentation and context:

Conjectures 1.3 as stated has a free parameter \delta: It says something along the lines that if the problem is hard for
degree n^{\delta} polynomials then it takes \exp(n^{\delta}) time.
However, I believe that the Theorems actually assuming the conclusion of the conjecture for every \delta in [0,1].
It is very important to clarify this as the starting point of the paper is not the MW conjecture but the outcome of this conditional conjecture assuming low-degree hardness for testing block model in exponential time.

I believe that the paper "Weak recovery, hypothesis testing, and mutual information in stochastic block models and planted factor graphs" by Mossel, Sly and Sohn show similar reductions for information theory variants of the problem. The authors should compare the results (and possibly proofs) to this setting.

There is no discussion of when is the interval below the KS bound where recovery is possible non-empty especially as the results are stated for parameters < 0.99 KS bound.

Some relevant papers here are:

Exact phase transitions for stochastic block models and reconstruction on trees by Mossel, Sly and Sohn as well as the more recent:

https://arxiv.org/abs/2503.03047 which discusses the case of multiple communities.

---

> ### Author Rebuttal · Authors · 2025-07-28
>
> Thank you for your constructive feedback and pointing out related literature!
>
> 0. Thank you for pointing out the ambiguity in the conjecture statement! You are absolutely right that conjecture 1.3 should be stated for every $\delta$ in [0,1], and our proof exploits conjecture 1.3 via the parameter $\delta$. We will make this more clear in the final version of the paper.
>
> 1. The hard interval can be empty when the number of communities k=2,3,4 (where there is no information-computation gap), or, if the conjecture stated in the paper is false and there are sub-exponential time algorithms whenever weak recovery is information-theoretically possible.
>
> 2. Our proof relies on the fact that the weak recovery estimator and the graph for cross validation are almost independent. Without the graph splitting procedure, the weak recovery estimator and the graph edges can be arbitrarily correlated. As a result, the testing statistics under null distribution can be very large, since the inner product of the centered adjacency matrix and the estimator can be very large. In particular, the almost independence is required to upper bound the inner product between the centered adjacency matrix and the weak recovery estimator with Bernstein inequality.

---

> > ### Comment · Reviewer_owhx · 2025-08-04
> >
> > Thanks for the replies.
> > Please clarify points 0. and 2. when you revise the paper.
> > Also for point 1. it would be good if you can make a positive statement - e.g. if the conjecture of the paper is true then the interval is not empty when k \geq 5 or k \geq sufficiently large constant (I think the later statement should be true).
> >
> > Also please add a reference and a discussion regarding the concurrent paper: Sharp Phase Transitions in Estimation with Low-Degree Polynomials by Sohn and Wein,

---

> ### Author Response · Authors · 2025-08-04
>
> We will keep it in mind, and incorporate into the future versions of our paper. For the relevant literature, we will cite and compare in the future versions of our paper. Thanks again for the constructive comments.
>
> For the concurrent paper: Sharp Phase Transitions in Estimation with Low-Degree Polynomials by Sohn and Wein [SW25], if we are not confused, it is already cited and discussed in details in section 2.1(the concurrent work paragraph) of our submission.

---

### Official Review · Reviewer_4FGD · 2025-06-28

**Clarity:** 4
**Significance:** 4
**Originality:** 3
**Rating:** 5
**Confidence:** 2

**Summary:**

This paper identifies several implications of the (extended) low-degree conjecture for the sparse SBM, recently put forward by Moitra and Wein (2023). The first implication (Theorem 2.1) is that efficient weak recovery below the Kesten-Stigum threshold is impossible. The second implication (Theorem 2.4) is a computational lower bound for learning the parameters of the SBM at a rate that is markedly better than the (suboptimal) SVD-based algorithm. Finally, Theorem 2.7 provides a computational lower bound for the related block graphon learning problem.

The proofs proceed by contradiction. For example, the proof of Theorem 2.1 begins by assuming a polynomial-time algorithm achieving weak recovery. The proof then constructs a function f which contradicts the assumption of Conjecture 1.3.

**Questions:**

N/A

**Ethical Concerns:**

["NO or VERY MINOR ethics concerns only"]

**Final Justification:**

I maintain my positive view of this paper, as the authors adequately responded to the questions raised.

**Limitations:**

yes

**Quality:**

4

**Strengths And Weaknesses:**

The area of computations/statistical thresholds has received significant interest in recent years. Few results rule out the possibility of efficient algorithms- the current state-of-the-art is to rule out e.g. algorithms based on low-degree polynomials. While the paper relies on a conjecture for its results, I feel this is nevertheless a strong contribution. Indeed, there are many works that rely on the Planted Clique Conjecture.

---

> ### Author Rebuttal · Authors · 2025-07-28
>
> Thank you for your review and constructive feedback!

---

> > ### Comment · Reviewer_4FGD · 2025-08-04
> >
> > I feel that the authors have adequately responded to the questions of the reviewers.

---

### Official Review · Reviewer_91pb · 2025-07-01

**Clarity:** 4
**Significance:** 3
**Originality:** 4
**Rating:** 5
**Confidence:** 3

**Summary:**

This work studies the computational hardness for weak recovery in Stochastic Block Models (SBMs). It has been conjectured that the Kesten-Stigum (KS) threshold is the computational efficiency threshold for beating the random guess on community labels on SBM. The main result of this paper states that for a slowly growing number of communities, no sub-exponential time algorithm can achieve weak recovery assuming a low-degree conjecture. This computational hardness result also implies computational lower bounds for learning the edge connection probability matrix of symmetric SBMs and learning block graphon function. These results establish interesting computational-statistical gaps with solid contributions to the literature.

**Questions:**

1. I notice a very recent paper on the same topic, by Chin, Mossel, Sohn, and Wein [1]. They establish the low-degree polynomial hardness result on weak recovery for $k \ll \sqrt{n}$. Do you believe the techniques used in your submission can potentially extend Theorem 2.2 to a wider regime on $k$, or we can actually find sub-exponential algorithms succeed on weak recovery when $n^{o(1)} \ll k\ll \sqrt{n}$?

2. Is the exponential of the rate $n^{-0.49}$ kind of arbitrary depending on the specific constants you choose in the analysis, as long as it is negative?

**Ethical Concerns:**

["NO or VERY MINOR ethics concerns only"]

**Final Justification:**

My questions have been resolved by the authors' responses. Those points were relatively minor and my review remains very positive. I keep my score as accept.

**Limitations:**

Yes.

**Paper Formatting Concerns:**

A minor point: the font style seems different from the one in the default NeurIPS format.

**Quality:**

4

**Strengths And Weaknesses:**

**Strengths**

1. This is one of the recent breakthroughs on providing rigorous evidence [1,2] for a sharp phase transition at the KS threshold.
2. Although the main result is not an unconditional lower bound on the weak recovery, the assumption on the low-degree conjecture makes it possible to rule out all algorithms run in time $\exp(n^{0.99})$.
3. The computational lower bounds on learning SBM are either new or better (in the regimes discussed in this paper) than prior works.
4. The writing is very clear, well-organized, and easy to follow.

**Weaknesses**

The techniques used in this paper seemingly limit the generalization of the results to a faster growing $k$.


---
Reference

[1] Sohn, Youngtak, and Alexander S. Wein. "Sharp phase transitions in estimation with low-degree polynomials." Proceedings of the 57th Annual ACM Symposium on Theory of Computing. 2025.

[2] Chin, Byron, et al. "Stochastic block models with many communities and the Kesten--Stigum bound." arXiv preprint arXiv:2503.03047 (2025).

---

> ### Author Rebuttal · Authors · 2025-07-28
>
> Thank you for your constructive feedback and pointing out related literature!
>
> 1. This is a very good question! It is not clear to us whether sub-exponential time algorithms exist in this regime. For hardness, we would expect that significantly new techniques are required to prove low degree lower bounds in this regime.
>
> 2. We can get lower bound for recovery rate $n^{-0.5+\delta}$ for any small constant $\delta$. We cannot get lower bound below the rate $n^{-0.5}$ , as such rate is achieved by algorithms which output random labelling. So our lower bound is essentially tight with respect to the rate.

---

> > ### Comment · Reviewer_91pb · 2025-08-04
> >
> > Thanks for the replies. I have no further questions.

---

### Official Review · Reviewer_ZnPh · 2025-07-02

**Clarity:** 4
**Significance:** 3
**Originality:** 4
**Rating:** 5
**Confidence:** 4

**Summary:**

This paper deduces strong lower bounds against recovery (estimation) in the sparse stochastic block model from a precise and strengthened version of the low-degree conjecture proposed in the recent work [MW23]. In particular, the authors' main result shows that, if there existed an estimation algorithm that estimated the matrix of community memberships (more precisely, the matrix whose entries show which nodes belong to the same community) even slightly better than a random guess of the partition into communities, for parameters slightly away from the boundary of the regime where the problem is conjectured to be computationally hard, then the conjecture formulated by [MW23] would be false. That conjecture is quite far from directly implying such a statement, so a rather subtle construction is required to translate the conjecture into this result. The authors also include some ancillary lower bounds that this implies for learning the parameters (in particular, the connection probabilities) of a stochastic block model, in various metrics.

**Questions:**

- I would like to see some more discussion of how one might arrive at the cross-validation idea. What goes wrong if one does not partition the given network into essentially two independent samples correlated with the same community structure? I assume that on a technical level it is this independence that plays an important role, but it is hard to pinpoint where that is so important.

- If I understand correctly, the authors' construction of the function f(Y) involves not just the input graph Y, but also some ancillary randomness, in order to perform the subsampling involved in the cross-validation construction. So, in order for the argument to go through, Conjecture 1.3 should also allow for such randomized algorithms f? (As an aside, it is an interesting question whether this extra randomness is necessary or not, and while sometimes papers about low-degree algorithms mention that randomized algorithms might equally well be allowed in formulations of such conjectures, I am not aware of a situation where it has played such an important role before.)

- The discussion of the concurrent work [SW25] is a little confusing---the setups and conclusions are rather different, because that work directly considers how well low-degree polynomial algorithms can perform estimation, while this work considers how well arbitrary algorithms can perform, conditional on the conjecture of [MW23]. Could the authors clarify this point? Perhaps mentioning the "low-degree MMSE" that [SW25] analyze might be helpful.

**Ethical Concerns:**

["NO or VERY MINOR ethics concerns only"]

**Final Justification:**

I did not raise any substantial issues but just a few clarifying questions, and the authors replied to my satisfaction. I am leaving my positive rating of the paper unchanged.

**Limitations:**

Yes.

**Paper Formatting Concerns:**

N/A.

**Quality:**

3

**Strengths And Weaknesses:**

Strengths: The main result is interesting and non-trivial to prove, coming from a clever construction of a function giving a good hypothesis test (in a certain precise "$L^2$ sense" adapted to the low-degree polynomials analysis) starting from a function estimating the community assignment. The authors' construction is based on cross-validating a given estimator, an idea I found intuitive but surprising, and that seems entirely original to this work as far as I can tell. I think this is a worthwhile contribution that deserves some attention and further study. The paper is also well-written and does a good job surveying related work and explaining the significance of the main result, which is rather technical-looking and might otherwise be difficult to contextualize for a reader who is not very familiar with either stochastic block models or low-degree algorithms.

Weaknesses: As far as I can tell, the above construction of a testing function is the only substantial theoretical innovation in the paper. The rest of the argument follows fairly directly with straightforward probability tools and one important tool from related literature (the correlation-preserving projection from the early work of Hopkins and Steurer) that is used in a black-box fashion. Relatedly, the main hardness result differs from previous similar results mostly in the details: what probability of success can an algorithm have, precisely how much estimation error can it make, etc. Thus it is a bit "in the weeds" of the stochastic block model and maybe of more interest to specialists. But, counterbalancing that, this style of proof has not been used before for the stochastic block model (or generally for discrete problems, to my knowledge), so I think the introduction of this argument together with the cross-validation construction may well turn out to inspire some follow-up work.

---

> ### Author Rebuttal · Authors · 2025-07-28
>
> Thank you for your review and constructive feedback!
>
> 1. Our proof relies on the fact that the weak recovery estimator and the graph for cross validation are almost independent. Without the graph splitting procedure, the weak recovery estimator and the graph edges can be arbitrarily correlated. As a result, the testing statistics under null distribution can be very large since the inner product of the centered adjacency matrix and the estimator can be very large. Particularly, the almost independence is required to upper bound the inner product of the centered adjacency matrix and the weak recovery estimator with Bernstein inequality.
>
> 2. Yes, conjecture 1.3 should allow randomized algorithms, we will clarify this in the final version of the paper. We totally agree that derandomizing our sample splitting algorithm is an interesting open problem.
>
> 3. Indeed, [SW25] directly provides unconditional lower bound on the low-degree MMSE, while our results rely on the low-degree conjecture. We list it as concurrent work, as both work provides computational lower bounds for weak recovery below the sharp Kesten-Stigum threshold, under the low-degree framework.

---

> ### Comment · Reviewer_ZnPh · 2025-08-04
>
> Thanks to the authors for their careful replies to my questions. My opinion of the paper remains very positive, and I have no further requests or questions.

---

### Decision · Program_Chairs · 2025-09-17

**Decision:**

Accept (spotlight)

**Comment:**

The paper gives a computational lower bound on weak recovery of community labels in a symmetric stochatic block model, as common in this literature by reducing to a specific conjecture.

The reviewer opinions are so unanimously positive that there is not much left to say. Although the topic is arguably not machine learning per se, it is a strong result in an area that is broadly of interest to machine learning. This is a clear accept.